# Incremental Learning of Sparse Attention Patterns in Transformers

**Oğuz Kaan Yüksel** [1]

## Abstract

This paper studies simple transformers trained on a high-order Markov chain, where the model must incorporate information from multiple past positions, each with different statistical importance. We show that transformers learn the task incrementally, with each stage corresponding to learning how to copy information from a subset of positions via a sparse attention pattern. Notably, the learning dynamics transition from a competitive phase, where all heads focus on the statistically most important positions, to a cooperative phase, where different heads specialize in different patterns. We model these dynamics with simplified differential equations and prove stage-wise convergence of the resulting system. Functionally, these stages correspond to a sequence of increasingly expressive misspecified models, with the full model class reached only at the end. Overall, we give a theoretical account of how structured attention patterns and head specialization emerge in stages without an explicit curriculum, with implications for generalization in sequential tasks.

## 1. Introduction

Transformers often acquire structure during training in stages: simple patterns appear first, and more complex behaviors emerge only later. Such stage-wise behavior has been observed in sequential tasks and language-modeling settings, including sudden capability gains and the progressive formation of attention circuits (Boix-Adsera et al., 2023; Edelman et al., 2024; Chen et al., 2024a). However, even in simple settings, it remains unclear how individual attention heads coordinate. Do heads *specialize* in different patterns from the start, or do they first *compete* for the same statistically dominant signal before differentiating? Understanding this mechanism is important for explaining how structured

attention patterns emerge from standard training dynamics.

A core primitive behind such structure formation is *copying*: moving information from one position to another so that it can be used in downstream computation. Copying is central in language modeling, where models must reuse phrases, entities, and contextual information, and in algorithmic reasoning, where intermediate values often need to be duplicated for use in later computations. Transformers implement copying through sparse attention patterns that concentrate probability mass on selected past positions; the simplest such pattern attends to a single position and is a subcircuit of *induction heads* (Elhage et al., 2021; Olsson et al., 2022). Exact sparsity requires the attention logits to separate in scale between attended and unattended positions. Thus, sparse copying is not only a representational primitive but also a *dynamical* one: the pressure to concentrate attention shapes how attention circuits form during training.

In this paper, we examine this mechanism in a controlled high-order Markov-chain task, illustrated in Figure 1. The next token depends on several groups of past positions, and each group is processed by a feature matrix with different statistical importance. The model must therefore learn multiple copying patterns and combine the copied information to predict the next token. We find that single-block decoder transformers learn this task incrementally. At first, all heads focus on the most statistically important positions, producing a competitive phase in which the heads learn the same sparse attention pattern instead of immediately dividing labor. Only later do the heads specialize: one *offshooting* head leaves the shared pattern and begins to copy from the next most important positions, while the remaining heads compensate for the residual contribution of the offshooting head. This competitive-to-cooperative transition is the central phenomenon of the paper. It shows that staged learning is not only visible in model performance, but also in how attention heads coordinate and specialize during training.

To explain this behavior, we introduce a simplified regression variant of the task and analyze its population gradient-flow dynamics. Under this simplification, the training dynamics reduce to nonlinear tensor factorization, a well-studied class of nonconvex problems (Arora et al., 2019; Razin et al., 2021; Li et al., 2021; Jin et al., 2023). Our theoretical results give sufficient conditions under which

[1]TML Lab, EPFL, Switzerland. Correspondence to: Oğuz Kaan Yüksel <oguz.yuksel@epfl.ch>.

*Proceedings of the 43rd International Conference on Machine Learning*, Seoul, South Korea. PMLR 306, 2026. Copyright 2026 by the author(s).

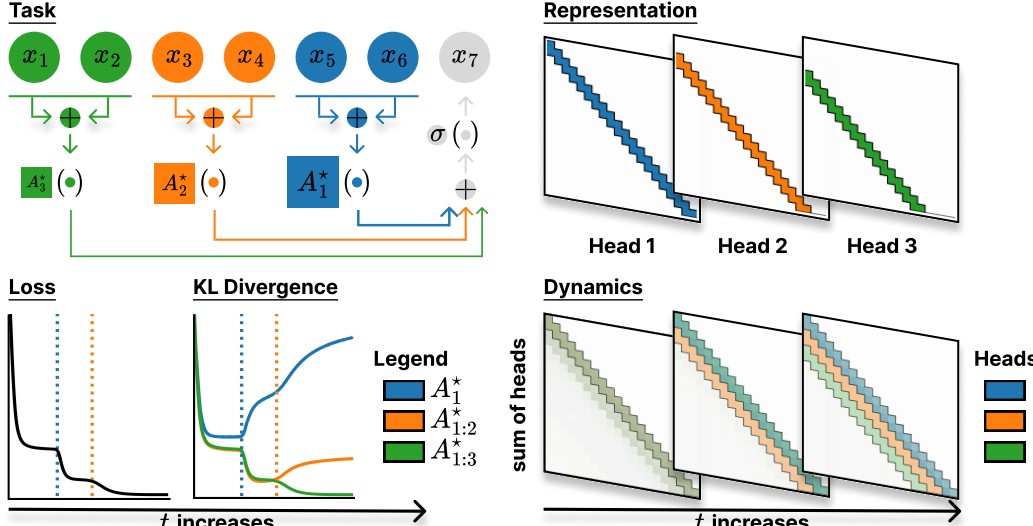

*Figure 1.* (Top left) The task is based on a high-order Markov chain, where the next token depends on multiple past tokens with different importance weights. The context is divided into groups of positions, each aggregated and mapped to prediction logits by an associated feature matrix $A_k^\star \in \mathbb{R}^{d \times d}$, with larger $\|A_k^\star\|$ indicating greater influence on the output. (Top right) An idealized representation of the task in a multi-head single-layer attention model. Each head represents an individual sparse attention pattern required to solve the task. (Bottom left) Transformers learn the task incrementally, with each stage corresponding to the acquisition of a sparse attention pattern, as indicated by the KL divergence between the transformer and predictors $A_{1:i}^\star$ that depend only on a prefix of the importance-ordered groups, as defined in Equation (2). (Bottom right) The learning dynamics transition from competitive, where all heads focus on the statistically most important pattern, to cooperative, where different heads specialize in different patterns.

near-symmetric initialization induces *coupled dynamics*: all heads converge toward the same dominant sparse pattern during the competitive phase. We then analyze how, near this coupled state, an offshooting head can move toward a new pattern while the other heads compensate, yielding the cooperative phase. Together, these results connect the observed head-level specialization in transformers to saddle-to-saddle dynamics. Our main contributions are as follows:

- **A simple task for positional incremental learning.** We introduce a high-order Markov-chain task that highlights sparse attention formation as a key mechanism for staged learning. Unlike prior in-context Markov-chain settings with more intricate multi-layer circuits (Edelman et al., 2024), the staged dynamics already appear here with a single self-attention layer.

- **Competitive-to-cooperative head dynamics.** Transformers first enter a competitive phase, where all heads focus on the statistically most important positions, before transitioning to a cooperative phase, where different heads specialize in different patterns. This phenomenon appears across variations in the importance ordering, within-group attention weights, interval overlap, initialization scale, optimizer, and dataset size.

- **A gradient-flow account of the stages.** We analyze a simplified regression model in which near-symmetric heads evolve as a coupled system, jointly converging toward the same dominant pattern during the competi-

tive phase. The analysis also captures the subsequent cooperative phase, where an offshooting head learns a new pattern while the remaining heads compensate.

- **Dataset size and effective complexity.** In low-data regimes, early-stopped models approach simpler effective predictors that use only the most important positions, suggesting a link between stage-wise learning and the effective complexity selected during training.

## 2. Stage-wise Formation of Sparse Patterns

In this section, we describe the data generation process, how transformers can represent the solution, the experimental evidence for incremental learning of sparse attention patterns, and how these dynamics vary across settings.

### 2.1. Markov Chains with Importance Structure

We consider a next-token classification task based on a discrete Markov chain of order $w$ over a dictionary $\mathcal{D}$ with $|\mathcal{D}| = d$. Each token in $\mathcal{D}$ is represented by a standard basis vector in $\mathbb{R}^d$. The initial tokens $x_{-w+1}, \ldots, x_0$ are sampled independently and uniformly from $\mathcal{D}$. For each $t \in [T]$, the remaining tokens are generated according to

$$x_t \sim \mathrm{softmax}\left(\sum_{k=1}^{h} A_k^\star \sum_{i \in I(k)} \alpha_i x_{t-i}\right), \qquad (1)$$

where $A_k^\star \in \mathbb{R}^{d \times d}$ are fixed feature matrices, $I(k)$ are disjoint subsets that partition $\{1, \ldots, w\}$, and $\alpha_i$ are nonnegative importance weights satisfying $\sum_{i \in I(k)} \alpha_i = 1$ for all $k \in [h]$. This task is simple but captures three features relevant to sequence modeling: (i) it is sequential, requiring the model to integrate information from past positions, (ii) it has positional structure, with the past lags partitioned into distinct groups, and (iii) different positions can have different importance, as determined by the feature matrices $A_k^\star$ and scalars $\alpha_i$. As $I(k)$ and $A_k^\star$ can be permuted without changing the data generation process, we assume without loss of generality that $\|A_1^\star\| \geq \|A_2^\star\| \geq \ldots \geq \|A_h^\star\|$ so that $I(1)$ is paired with the most influential feature matrix.

One particular choice of interest is to have $I(k)$ be contiguous blocks of indices that start from the most recent position, i.e., for some $0 = i_0 < i_1 < i_2 < \ldots < i_{h-1} < i_h = w$ and $I(k) = \{i_{k-1} + 1, \ldots, i_k\}$. Contiguous blocks serve as one illustrative default, motivated by the stronger statistical correlations between nearby tokens in natural language. Notably, when each $I(k)$ is a singleton, the task recovers position-wise copying followed by a linear feature map. Lastly, the task is not *identifiable*: the sum in Equation (1) admits multiple decompositions with different feature matrices $A_k^\star$ and partitions $I(k)$; see Section C for examples.

## 2.2. Representing the Task with Sparse Attention

We construct a single-layer multi-head attention model that solves the task. The construction assigns one head to each group: the attention pattern copies the relevant past tokens, and the value matrix applies the corresponding feature map.

Let $X \in \mathbb{R}^{d \times (T+w)}$ be the input data matrix with columns $x_{-w+1}, \ldots, x_0, x_1, \ldots, x_T$. For a transparent construction, we encode positional information using one-hot vectors in $\mathbb{R}^{T+w}$ and concatenate them with the input as follows: [1]

$$\tilde{X} = \begin{pmatrix} X \\ I_{T+w} \end{pmatrix} \in \mathbb{R}^{(d+T+w) \times (T+w)}.$$

We denote the columns of $\tilde{X}$ as $\tilde{x}_i \in \mathbb{R}^{d+T+w}$, the position-augmented embedding of token $x_i$. The transformer takes $\tilde{X}$ as input and produces the output $Y \in \mathbb{R}^{d \times T}$ with columns $y_0, \ldots, y_{T-1}$ as follows:

$$y_t = \text{softmax}\left(\sum_{k=1}^{h} V_k \tilde{X} a_t^{(k)}\right),$$

$$a_t^{(k)} = \text{softmax}\left(\mathcal{M}_{T-t}\left(\tilde{X}^\top K_k^\top Q_k \tilde{x}_t\right)\right),$$

where $Q_k, K_k \in \mathbb{R}^{(d+T+w) \times (d+T+w)}$ are the query and key matrices of head $k$, $V_k \in \mathbb{R}^{d \times (d+T+w)}$ is the value

matrix, and $\mathcal{M}_p$ sets the last $p$ entries to $-\infty$ to apply causal masking.

For head $k$, we let $a_t^{(k),\star}$ denote the ideal positional attention pattern corresponding to $I(k)$, defined entrywise by

$$\left(a_t^{(k),\star}\right)_s = \begin{cases} \alpha_i, & \text{if } s = t - i \text{ for some } i \in I(k), \\ 0, & \text{otherwise}, \end{cases}$$

where $s$ ranges over the positions in the input sequence. We choose $V_k$ to apply $A_k^\star$ to the token coordinates and discard the positional coordinates, i.e., $V_k = [A_k^\star \ 0]$. With this choice,

$$V_k \tilde{X} a_t^{(k),\star} = A_k^\star \sum_{i \in I(k)} \alpha_i x_{t-i},$$

so each head recovers one contribution to the logits in Equation (1). By *sparse attention*, we mean the limiting case in which attention mass is concentrated on a subset of positions and vanishes on the complement. Exactly learning such a pattern requires the logit gap between attended and unattended positions to diverge. In practice, we expect finite logit separations that approximate these sparse attention patterns. This sparse pattern can be realized using only positional coordinates by setting

$$K_k^\top Q_k = \lambda \sum_{i \in I(k)} \sum_{p=w}^{T+w} e_{d+p-i} e_{d+p}^\top,$$

where $\lambda > 0$ is a scaling constant and $e_i$ is the $i$-th standard basis vector in $\mathbb{R}^{d+T+w}$. As $\lambda \to \infty$, the attention scores converge to the desired sparse pattern.

This construction is not unique as there are many $Q_k$ and $K_k$ that can realize the same attention pattern. In particular, there is a symmetry in which $(Q_k, K_k)$ can be replaced with $\left(M^{-1} Q_k, M^\top K_k\right)$ for any invertible matrix $M$ without changing the attention scores. Moreover, as there are $h$ heads to learn, the construction has a permutation symmetry. The permutation symmetry plays a central role in our analysis of the learning dynamics, as we discuss in Section 3.2.

## 2.3. Transformers Learn Incrementally

We train single-block decoder transformers with $h$ heads on sequences sampled from Equation (1), minimizing cross-entropy loss over the full sequence. The initial tokens $x_{-w+1}, \ldots, x_0$ are excluded since they are not drawn from the process. We keep the architecture as close to standard practice as possible. The architecture and optimization details are provided in Section A.

We sample feature matrices $A_k^\star$ uniformly from orthogonal matrices and then scale them by positive scalars $m_k$. These constants are chosen geometrically, i.e., $m_k = m^{h-k} b_0$ where $m > 1$ is the multiplicative constant and $b_0 > 0$ is the

---

[1] We use one-hot positional encodings for clarity. The construction and experiments are compatible with relative or sinusoidal encodings that avoid $\mathcal{O}(T)$ embedding dimension.

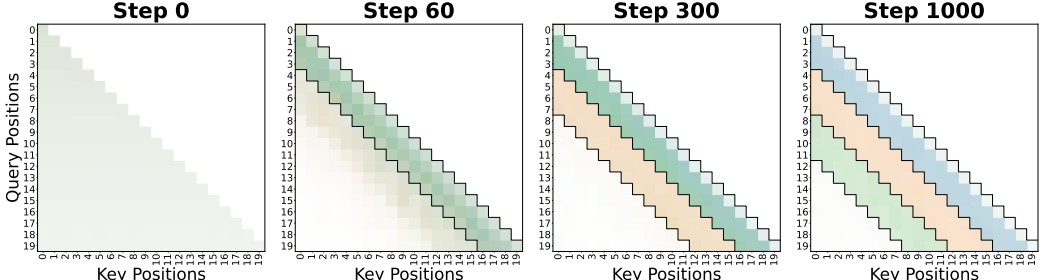

*Figure 2.* The sum of learned attention patterns for $h = 3, w = 12$ at different stages of training with blue, yellow, and green corresponding to different heads. At $t = 0$, the attention is uniform as the model is randomly initialized. At $t = 60$, all heads learn from the positions in $I(1)$, indicated by the overlapping blue, yellow, and green, with some residual attention also placed on the positions in $I(2)$. At $t = 300$, a head learns from the positions in $I(2)$ whereas two heads still focus on $I(1)$. At $t = 1000$, the model finally learns to integrate all positions, with each head specializing in a different pattern. The main diagonal does not have the same intensity as the other positions, as it is learned directly from the input via the skip connection.

base scale. This results in an importance hierarchy across the feature matrices, while features within the same matrix are equally important. In particular, $A_1^\star$ has the largest norm and thus contains the most influential features, while $A_h^\star$ has the smallest norm and contains the least important ones. For simplicity, we choose $I(k)$ as in Section 2.1 with equal-length intervals of size $w/h$, and set $\alpha_i = 1/|I(k)|$ for $i \in I(k)$. See Section A for experimental details and Section B for additional experiments.

We observe that the transformers learn the task incrementally, with each stage corresponding to the acquisition of a sparse attention pattern as in Figure 2. At initialization, all heads have uniform attention. Early in training, they focus primarily on $I(1)$, the most statistically important positions. This is the *competitive phase*: the heads compete to learn from these positions, resulting in overlapping attention patterns with initialization-induced deviations. The dynamics then enter the *cooperative phase*, where heads gradually specialize in different patterns, with one head learning from $I(2)$ while another eventually focuses on $I(3)$. Notably, training consistently recovers the sparse, disjoint representation, even though the task admits multiple decompositions.

To probe these dynamics in function space, we train transformers with restricted maximum context lengths $c = 4, 8, 12$. With $c = 4$, the model can only access $I(1)$; with $c = 8$, it can additionally access $I(2)$; with $c = 12$, it can access all relevant positions and solve the task fully. In Figure 3 (right), we plot the Kullback-Leibler (KL) divergence between each restricted model and a full-context transformer over training. The full-context transformer first approaches the $c = 4$ model, then the $c = 8$ model, before approaching the $c = 12$ model. This indicates that the transformer jointly learns each attention pattern and its associated feature matrix at every stage, rather than acquiring them separately.

We obtain a similar picture by comparing the transformer's

predictions to ground-truth predictors that depend only on the positions in $I(1)$, $I(1) \cup I(2)$, and $I(1) \cup I(2) \cup I(3)$:

$$f_{A_{1:i}^\star} = \text{softmax}\left(\sum_{k=1}^{i} A_k^\star \sum_{j \in I(k)} \alpha_j x_{t-j}\right). \quad (2)$$

Figure 3 (left) shows the same incremental pattern: the transformer first approaches $f_{A_1^\star}$, then $f_{A_{1:2}^\star}$, before approaching $f_{A_{1:3}^\star}$. This mirrors what Edelman et al. (2024) observed: stages characterized by sub-$n$-grams.

### 2.4. Ablation Studies

To identify the architectural components needed for incremental learning, we remove elements absent from the idealized construction in Section 2.2, such as layer normalization and residual connections. We further reduce the product $K_k^\top Q_k$ to a single matrix $A_k$. Individually or in combination, these simplifications do not significantly alter the incremental dynamics; we plot the results in Figure 1.

We perform ablation studies with this minimal architecture. We first vary the initialization scale of the attention matrices $A_k$ and set value matrices to zero to isolate the attention dynamics. We initialize $A_k$ from a uniform distribution on $[-u, u]$, where $u$ is the initialization scale. Figure 4 (left) shows that the speed of incremental learning is affected by the initialization scale, with smaller scales resulting in slower learning. At the extreme $u = 0$, we observe that the model only learns a single pattern and does not progress further. The heads are exactly symmetric at $u = 0$, and breaking this symmetry requires a small perturbation.

We also vary the multiplicative constant $m$ that controls the importance hierarchy in the data generation process. Figure 4 (right) shows that the number of distinct stages decreases to two for $m = 1$, where there is no importance ordering. When $m = 1$, the model first learns one pattern and then acquires the remaining two simultaneously. For

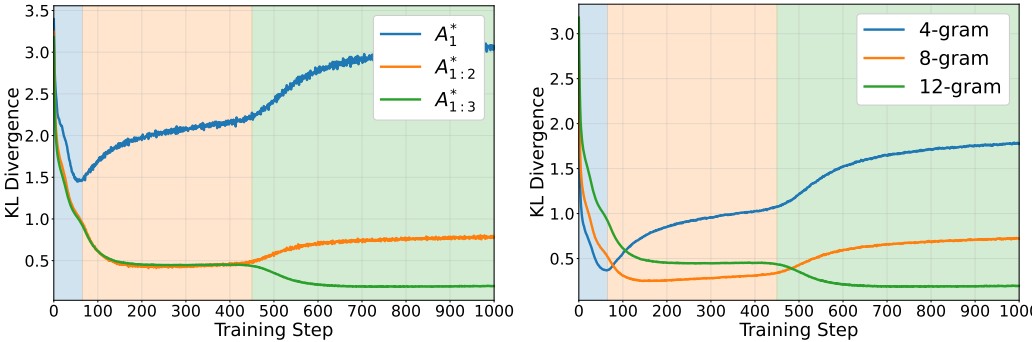

*Figure 3.* Stage-wise learning in function space: each curve traces the proximity of the full model to a predictor along the training trajectory. Curves to intermediate predictors dip and then rise as the trajectory passes through their neighborhood and incorporates additional positions, while the curve to the full predictor decreases and stays low. (Left) KL divergence between the ground-truth predictors that depend only on the positions in $I(1)$, $I(1) \cup I(2)$, and $I(1) \cup I(2) \cup I(3)$, and the predictions of the full-context transformer. (Right) KL divergence between the predictions of transformers with restricted context lengths $c = 4, 8, 12$, and those of the full-context transformer.

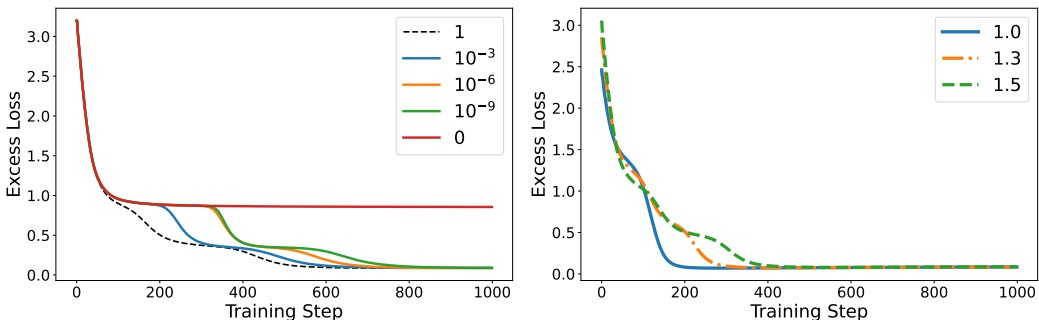

*Figure 4.* (Left) Excess loss of the minimal architecture with different initialization scales. Smaller scales yield slower learning; at $u = 0$, the heads remain exactly symmetric and only the first pattern is learned. (Right) Excess loss of the minimal architecture with different multiplicative constants $m$ that determine the importance hierarchy. When $m > 1$, the importance gap between features attracts the initialization to each saddle sequentially, yielding three distinct stages. When $m = 1$, there is no importance ordering, so the initialization is not preferentially attracted to any saddle; one pattern is learned first due to random symmetry breaking, and the remaining two are acquired simultaneously, collapsing to two stages.

$m = 1.3$ and $m = 1.5$, three distinct stages emerge, though more intertwined for $m = 1.3$ with less pronounced bumps in the loss curve. Overall, larger importance gaps yield more clearly separated stages; when the gaps shrink, multiple patterns can be acquired at nearly the same time.

Beyond initialization scale and the importance hierarchy, Section B reports ablations over reversed orderings, overlapping intervals, non-uniform $\alpha_i$, and additional heads. The competitive-to-cooperative dynamics appear across these variations, indicating that the staged behavior is not specific to the particular choices in our default setup. In less structured settings, such as overlapping intervals, there can be multiple sparse solutions, and the initialization and feature matrices can influence which solution is selected.

## 2.5. Dataset Size and Effective Complexity

Lastly, we study the effect of dataset size on incremental learning. As we decrease the dataset size below certain thresholds, the number of stages observed in training decreases, as shown in Figure 5 (left). Figure 5 (right) plots the KL divergence between restricted-context transformers and the low-data trained transformer. The trend is similar to that observed in Figure 3, but with fewer stage transitions as the dataset size decreases.

In low-data regimes, the early-stopped transformer approaches simpler effective predictors that depend only on the most important positions. This is consistent with a regularization effect associated with the training trajectory, where shorter effective context corresponds to a simpler, misspecified hypothesis class. Yüksel et al. (2025) argue that such misspecification can be beneficial in low-data regimes, improving sample efficiency. Transformers with early stopping seem to select the effective context length automatically, suggesting a possible sample-complexity benefit.

Notably, this structure appears to persist in the overfit regime: with 600 samples, for example, the model concen-

trates attention on the first two patterns and gives minimal weight to the third, even when the corresponding feature matrices overfit to training noise. This hierarchical allocation could explain why early stopping recovers a clean predictor on the top-$k$ learnable patterns.

## 3. Training Dynamics on Regression Variant

In this section, we introduce a tractable surrogate regression variant of the classification task from Section 2.1, using the minimal architecture of Section 2.4. We study its training dynamics by analyzing the population gradient flow.

### 3.1. The Regression Model

Consider the following regression task associated with any distribution $\mathcal{P}_X$ and $\mathcal{P}_\xi$: $(x_1, \ldots, x_T) \sim \mathcal{P}_X, \xi \sim \mathcal{P}_\xi$, and

$$y^\star(X) = \sum_{k=1}^h A_k^\star X s_k^\star + \xi \,,$$

where $s_k^\star \in \mathbb{R}^T$ is the vector with entries $\alpha_i$ for $i \in I(k)$ and zero otherwise. For this section, we set $|I(k)| = 1$ for all $k$ for simplicity. Let $m_k^\star = \|A_k^\star\|_F$ and $V_k^\star = A_k^\star/\|A_k^\star\|_F$ for all $k \in [h]$ with $m_1^\star > \ldots > m_h^\star$ without loss of generality.

We make some assumptions regarding the distributions $\mathcal{P}_X, \mathcal{P}_\xi$, and the feature matrices.

**Assumption 1.** *The noise is zero-mean, i.e., $\mathbb{E}[\xi] = 0$ and the data is normalized, i.e.,*

$$\forall i, j \in [T], \quad \mathbb{E}\left[x_i x_j^\top\right] = \mathbf{1}_{i=j} I_d \,.$$

Any isotropic covariance reduces to this case by absorbing the scale into the norms $m_k^\star$; e.g., random one-hot tokens.

**Assumption 2.** *The feature matrices are orthogonal, i.e.,*

$$\forall i, j \in [h], \quad \langle V_i^\star, V_j^\star \rangle = \mathrm{Tr}\left((V_i^\star)^\top V_j^\star\right) = \mathbf{1}_{i=j} \,.$$

These assumptions are made for analytical tractability. The isotropic covariance assumption turns the population objective into a pure tensor-factorization problem rather than a matrix-sensing problem, and the orthogonality assumption makes the feature ordering $m_1^\star > \cdots > m_h^\star$ unambiguous. Relaxing them would require replacing this scalar ordering with an ordering in the geometry induced by the data covariance and target tensor, which we leave outside the scope of the present analysis.

We use the minimal architecture obtained in Section 2.4 with the following modifications. The attention scores are computed only via the inner product of position vectors, instead of the concatenated position and data vectors. As the problem is a regression task on the final token, we only

need the last row of the matrix $Q_k$ which we denote by $q_k \in \mathbb{R}^T$. Then, the resulting model is as follows:

$$y_\theta(X) = \sum_{k=1}^h V_k X s_k \,, \quad \text{with} \quad s_k = \mathrm{softmax}(q_k) \,,$$

where $\theta = (V_1, \ldots, V_h, q_1, \ldots, q_h)$ are the learnable parameters. We set the loss to the mean-squared loss:

$$\mathcal{L}(\theta) = \frac{1}{2}\mathbb{E}_{x_1,\ldots,x_T,\xi}\left[\|y_\theta(X) - y^\star(X,\xi)\|^2\right] \,. \quad (3)$$

We study the gradient flow dynamics of the population loss in Equation (3), i.e., we consider the continuous-time limit of gradient descent with infinitesimal step size.

**Tensor Notation.** We construct tensors that are sums of outer products of matrices and vectors, i.e., $\boldsymbol{M} = \sum_{k=1}^h B_k \otimes v_k$, where $B_k \in \mathbb{R}^{d \times d}$ and $v_k \in \mathbb{R}^T$. The product $X^\top \boldsymbol{M}$ denotes $X^\top \boldsymbol{M} = \sum_{k=1}^h \langle B_k, X \rangle v_k$ whereas the product $\boldsymbol{M} v$ denotes $\boldsymbol{M} v = \sum_{k=1}^h B_k \langle v_k, v \rangle$. The inner product between two tensors $\boldsymbol{M} = \sum_{k=1}^h B_k \otimes v_k$ and $\boldsymbol{N} = \sum_{k=1}^h B_k' \otimes v_k'$ is denoted by $\langle \boldsymbol{M}, \boldsymbol{N} \rangle = \sum_{k=1}^h \langle B_k, B_k' \rangle \langle v_k, v_k' \rangle$. The Frobenius norm of a tensor $\boldsymbol{M}$ is given by $\|\boldsymbol{M}\|_F = \sqrt{\langle \boldsymbol{M}, \boldsymbol{M} \rangle}$.

Proposition 1 reinterprets this dynamical system as gradient flow for a tensor factorization problem.

**Proposition 1.** *The gradient flow dynamics of the loss in Equation (3) is equivalent to those of $\mathcal{L}(\theta) = \dfrac{1}{2}\|\boldsymbol{G} - \boldsymbol{P}\|_F^2$*

$$\text{where} \quad \boldsymbol{P} = \sum_{k=1}^h V_k \otimes s_k \,, \quad \text{and} \quad \boldsymbol{G} = \sum_{k=1}^h m_k^\star \left(V_k^\star \otimes s_k^\star\right) \,.$$

**Attention Reparameterization.** Note that due to the softmax operation, $\sum_i q_i$ is always constant, and thus we can restrict $q_k$ to have a zero mean without loss of generality. This implies that there is a one-to-one correspondence between $q_k$ and $s_k$ in the subspace of zero-mean vectors. Therefore, it is possible to analyze the dynamics in terms of $s_k$ instead of $q_k$ with the notation $\Pi(s) = \left(\mathrm{diag}(s) - ss^\top\right)$:

$$\dot{V}_k = (\boldsymbol{G} - \boldsymbol{P})s_k \,, \quad \dot{s}_k = \Pi(s_k)^2\left(V_k^\top(\boldsymbol{G} - \boldsymbol{P})\right) \,. \quad (4)$$

**Numerical Simulations.** We simulate these differential equations with initialization $V_i = 0$ and $s_i \approx \frac{1}{T}\mathbf{1}_T$. The results recapitulate the incremental learning behavior observed in Figure 2. We present the results in Section D.

### 3.2. Coupled Dynamics Describe the Competitive Phase

We analyze a sufficient idealized regime for the competitive phase: near-symmetric initialization induces coupled dynamics, with $s_1(0) = s_k(0), V_1(0) = V_k(0)$ for

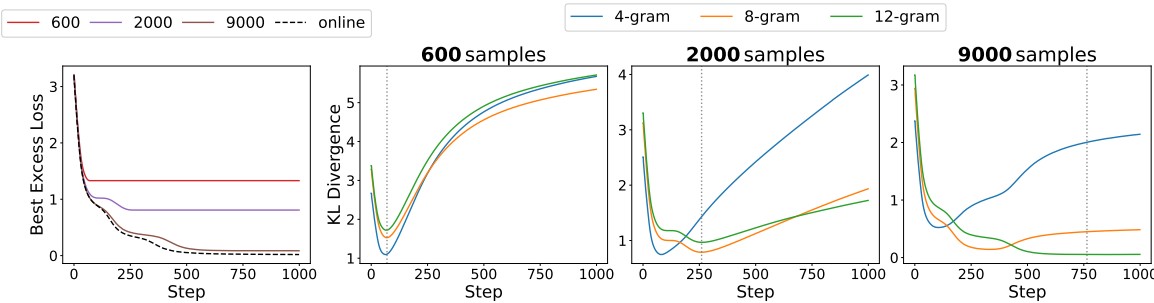

*Figure 5.* Smaller datasets reduce the number of stages reached during training. (Left) The best validation loss as a function of the dataset size. (Right) The KL divergence between restricted-context transformers and the trained transformer. Dashed lines indicate the first step that obtains the best excess loss.

all $k$. Once the heads are coupled, they co-evolve, i.e., $s_k(0) = s(0), V_k(0) = V(0)$ for all $k$.

This leads to the following coupled dynamics:

$$\dot{V} = \left(\boldsymbol{G}s - h\|s\|^2 V\right) , \quad \dot{s} = \Pi(s)^2 \left(V^\top \boldsymbol{G} - h\|V\|_F^2 s\right) .$$

**Theorem 1.** *Assume that the initialization satisfies the following for all $k \in [h]$:*

$$\langle V(0), V_1^\star \rangle \geq \langle V(0), V_k^\star \rangle , \quad \langle s(0), s_1^\star \rangle \geq \langle s(0), s_k^\star \rangle . \quad (5)$$

*Then, the dynamics of $V$ and $s$ converge to the following fixed point:*

$$V(\infty) = \frac{m_1^\star}{h} V_1^\star , \quad s(\infty) = s_1^\star . \quad (6)$$

Theorem 1 is based on an ordering argument. As long as the initialization satisfies the ordering condition in Equation (5), the dynamics of $V$ and $s$ are such that $\dot{V}$ and $\dot{s}$ reinforce the same order. On its own, Theorem 1 does not explain what happens when the heads do not start with the same initialization. Theorem 2 establishes that when many heads are initialized with a small deviation from the symmetric initialization, the deviation from the symmetric initialization is bounded for a finite time that we can precisely control. Therefore, the initialization influences the coupling time of different heads, after which they might start to diverge.

**Theorem 2.** *Assume that the following holds for $\epsilon \ll 1$:*

$$\forall k \in [h] : \|V(0) - V_k(0)\|_F \leq \epsilon \text{ and } \|s(0) - s_k(0)\|_2 \leq \epsilon .$$

*Then, there exists a constant $c_1$ such that $\forall t \in \left[0, \frac{1}{-c_1 \log \epsilon}\right]$:*

$$\|V_k(t) - V(t)\|_F \leq \epsilon e^{c_1 t} \quad \text{and} \quad \|s_k(t) - s(t)\|_2 \leq \epsilon e^{c_1 t} .$$

Lastly, we remark that the initialization in Theorem 1 can be further relaxed to a wider basin of attraction around the symmetric initialization of interest. This follows from a similar argument as in Zucchet et al. (2026) who has studied the escape time from this initialization when $h = 1$.

**Remark 1.** *The initialization of interest is $s_k(0) \approx \frac{1}{T} 1_T$ for all $k \in [h]$ as seen in Figure 2. By expanding the dynamics around this initialization with $V_k \approx 0$ for all $k \in [h]$:*

$$\dot{V}_k(0) \approx \frac{1}{T} \boldsymbol{G} 1_T , \quad \dot{s}_k(0) \approx 0 .$$

*Similarly, second-order local approximation shows that $s_k$ has the largest increase in the direction $s_1^\star$. Therefore, we can quantify a wider basin of attraction for Theorem 1 as all $V_k$ and $s_k$ move toward the initialization space defined by Equation (5).*

### 3.3. Cooperation After Competition

To study the cooperative phase after the initial competitive phase, we consider the dynamics of the loss at various initializations around the fixed point in Equation (6). Consider the following initialization scheme:

$$V_1(0) = \ldots = V_{h-1}(0) \approx \frac{m_1^\star}{h} V_1^\star , \quad V_h(0) \approx \frac{m_1^\star}{h} V_1^\star ,$$
$$s_1(0) = \ldots = s_{h-1}(0) = s_1^\star , \quad s_h(0) \approx s_1^\star . \quad (7)$$

The dynamics of $s_1, \ldots, s_{h-1}$ remain fixed because the projection term vanishes at $s_1^\star$, i.e., $\Pi(s_k) = 0$ for all $k \in [h-1]$. In addition, $V_1, \ldots, V_{h-1}$ are coupled due to the gradient flow. Therefore, the whole system collapses to three equations: one for $V$ that describes the ensemble and two for $V'$ and $s'$ that describe the offshooting head:

$$\dot{s}' = \Pi(s')^2 \left(V'^\top \boldsymbol{G} - (h-1)\langle V', V \rangle s_1^\star - \|V'\|^2 s'\right) ,$$
$$\dot{V} = m_1^\star \|s_1^\star\|^2 V_1^\star - (h-1)\|s_1^\star\|^2 V - \langle s_1^\star, s' \rangle V' ,$$
$$\dot{V}' = \boldsymbol{G}s' - (h-1)\langle s_1^\star, s' \rangle V - \|s'\|^2 V' , \quad (8)$$

We have a similar control to Theorem 2 for the dynamics of $V, V'$, and $s'$. Theorem 3 establishes that the deviation from the cooperative system is bounded for a finite time that we can precisely control. This is due to a Lyapunov control argument in which the norms of $V$ and $V'$ are bounded.

**Theorem 3.** *Assume that the following holds for $\epsilon \ll 1$:*

$$\forall k \in [h-1] : \|V(0) - V_k(0)\|_F \leq \epsilon, \|s_1^\star - s_k(0)\|_2 \leq \epsilon,$$
$$\text{and} \quad \|V'(0) - V_h(0)\|_F \leq \epsilon, \|s'(0) - s_h(0)\|_2 \leq \epsilon.$$

*Let $\Delta(t)$ be the deviation from the cooperative system in Equation (8):*

$$\Delta(t) = \max \Big\{ \max_{k \in [h-1]} \{\|V_k(t) - V(t)\|_F, \|s_k(t) - s(t)\|_2\},$$
$$\|V_h(t) - V'(t)\|_F, \|s_h(t) - s'(t)\|_2 \Big\}.$$

*Assuming that $\|s'(t) - s_1^\star\| \geq \delta$ for all $t \in \mathbb{R}$, there exists a universal constant $c_1$ such that:*

$$\Delta(t) \leq \epsilon e^{c_1 t}, \quad \forall t \in \left[0, \frac{1}{-c_1 \log \epsilon}\right].$$

The dynamics in Equation (8) with the initialization in Equation (7) are nontrivial: while $V'$ grows in an orthogonal direction $V_\perp$ to $V_1^\star$, $s'$ is still sparse around $s_1^\star$. This is due to the fact that $\Pi(s') \approx 0$ at initialization as $s'(0) \approx s_1^\star$ which leads to a scale separation between $\dot{s}'$ and $\dot{V}'$. Consequently, when $V'$ grows along some $V_\perp$, the prediction is pushed to include the unnecessary term, $V_\perp x_t$. However, this is immediately canceled out by the progression of the ensemble, where $V$ learns to offset this by learning $-V_\perp$.

This collaborative behavior is best seen in Figure 19. We also verify that the same compensatory behavior in the feature matrices appears in the minimal transformer trained on the classification task in Figure 21, indicating that our theory captures non-trivial structural aspects of the dynamics beyond the existence of stages. This comparison is qualitative: the two settings differ in loss, parameterization, and timescale, so we compare structural signatures rather than aligned numerical trajectories.

To simplify Equation (8), we show that the initialization in Equation (7) ensures that $V$ is close to its optimal value, $V^\star$, which is defined in Lemma 1. In fact, we can derive a precise statement about how far $V$ is from $V^\star$ based on the weight $s'$ puts on the direction of $s_1^\star$:

**Lemma 1.** *Let $\Delta(t) = V(t) - V^\star(t)$ where*

$$V^\star(t) = \frac{1}{h-1} \left(m_1^\star V_1^\star - \langle s_1^\star, s'(t)\rangle V'(t)\right).$$

*Assuming that $\|s'(t) - s_1^\star\| \geq \delta$ for all $t \in \mathbb{R}$, there exist constants $c_1(\delta), c_2$ such that*

$$\|\Delta(t)\|_F \leq e^{-c_2 t}\|\Delta(0)\|_F + \frac{c_1(\delta)}{c_2}.$$

Inspired by Lemma 1 and numerical simulations in Figure 20, we use a two-scale approximation in which $V$ is optimized faster:

$$\dot{V}' = \boldsymbol{G}_{(1)} s'_{(1)} - \|s'_{(1)}\|^2 V',$$
$$\dot{s}' = \Pi(s')^2 \left(V_{(1)}'^\top \boldsymbol{G}_{(1)} - \|V'\|_F^2 s'_{(1)}\right), \quad (9)$$

where we introduce $\boldsymbol{G}_{(i)} = \boldsymbol{G} - \sum_{j=1}^i m_j^\star \left(V_j^\star \otimes s_j^\star\right)$ along with the following notation:

$$V_{(i)} = V - \sum_{j=1}^i \langle V_j^\star, V\rangle V_j^\star, \quad s_{(i)} = s - \sum_{j=1}^i \langle s_j^\star, s\rangle s_j^\star.$$

We show that the dynamics in Equation (9) converge to the second positional feature:

**Theorem 4.** *Assume that the initialization satisfies the following for all $k \in [2, h]$:*

$$\langle V'(0), V_2^\star\rangle \geq \langle V'(0), V_k^\star\rangle, \quad \langle s'(0), s_2^\star\rangle \geq \langle s'(0), s_k^\star\rangle.$$

*Further, suppose that $V'(0), s'(0)$ are such that*

$$\langle V_{(1)}'(0), \boldsymbol{G}_{(1)} s_{(1)}'(0)\rangle > \frac{1}{2}\|V'(0)\|_F^2 \|s_{(1)}'(0)\|^2. \quad (10)$$

*Then, the dynamics of $V'$ and $s'$ converge to the following fixed point:*

$$V'(\infty) = m_2^\star V_2^\star, \quad s'(\infty) = s_2^\star.$$

Theorem 4 is similar in nature to Theorem 1. Once there is alignment with the second positional feature, the dynamics are such that the alignment is not broken. Notably, we require the initialization to satisfy Equation (10). This ensures that the dynamics start with an initial decrease on the loss beyond the first saddle point characterized in Theorem 1: Theorem 4 proves that a potential characterizing the loss is monotonically minimized and this saddle is avoided. In Remark 2, we discuss how a small perturbation toward the second positional feature is sufficient to satisfy Equation (10).

Finally, in Section E.4, we extend the same argument to the specialization of an arbitrary head after the system has acquired the earlier features. In later phases, the previously specialized heads are treated as approximately fixed at their learned features, while the remaining coupled ensemble still represents the dominant feature and one free head offshoots toward the next feature. Together with Theorems 2 and 3, the analogous stability bounds show that each phase remains close to the corresponding idealized coupled dynamics for a finite, controlled time.

## 4. Related Work and Discussion

**Incremental learning.** Plateau-like learning curves are a common feature in neural network training. Early analyses, such as Fukumizu & Amari (2000), attributed these

behaviors to critical points in supervised learning. Subsequent studies have examined similar dynamics in a variety of simplified settings, including linear networks (Gissin et al., 2020; Saxe et al., 2019; Gidel et al., 2019; Arora et al., 2019; Jacot et al., 2021; Li et al., 2021; Razin et al., 2021; Jiang et al., 2023; Berthier, 2023; Pesme & Flammarion, 2023; Jin et al., 2023; Varre et al., 2023; 2024) and ReLU models (Boursier et al., 2022; Abbe et al., 2023); recent work argues for their universality (Ziyin et al., 2025; Kunin et al., 2025; Zhang et al., 2026). In transformer training, plateaus followed by sudden capability gains (Chen et al., 2024a) are often observed in regression tasks (Garg et al., 2022; von Oswald et al., 2023; Ahn et al., 2023) and formal language recognition (Bhattamishra et al., 2024; Akyürek et al., 2024; D'Angelo et al., 2025; Cagnetta et al., 2025; D'Angelo & Flammarion, 2026). Finally, Cagnetta & Wyart (2024); Cagnetta et al. (2025) show dataset size affects the order of the learned hierarchy in random probabilistic context-free grammars, mirroring our data-dependent stage progression and its connection to effective model complexity.

$n$**-gram models.** $n$-gram language models (Jurafsky & Martin, 2009) serve as a toy setting to understand language models. This perspective has motivated studies of optimization landscapes (Makkuva et al., 2025), expressivity over $n$-gram distributions (Svete & Cotterell, 2024), and sample complexity (Yüksel & Flammarion, 2025b;a). The learning of variable-order $n$-grams has been studied by Zhou et al. (2026), while Deora et al. (2025) consider $n$-grams of different fixed orders $n$. Connections between in-context learning and the emergence of induction heads (Elhage et al., 2021; Olsson et al., 2022), together with their acquisition via gradient descent (Nichani et al., 2024), are drawn by Bietti et al. (2023). Training dynamics on $n$-gram tasks have also been shown to progress in stages: intermediate solutions approximate sub-$n$-grams (Edelman et al., 2024), which are formalized as near-stationary points by Varre et al. (2025). Despite this rich phenomenology, standard $n$-gram models lack the hierarchical abstractions characteristic of natural language (Wu et al., 2022; 2025). Our task adds an importance hierarchy over positional features, while leaving richer forms of abstraction and composition outside its scope.

**Attention dynamics.** The dynamics of attention have recently been explored through various simplified tasks (Snell et al., 2021; Li et al., 2023; Yang et al., 2024; Huang et al., 2024; Goel et al., 2026), including tasks exhibiting sparsity (Jelassi et al., 2022; Tian et al., 2023; Wang et al., 2024). Particularly relevant to our task, Marion et al. (2025) study single-location regression, in which attention concentrates on a randomly selected informative position, a setting related to sequence multi-index models (Cui et al., 2024; Troiani et al., 2025). Tractable analysis usually requires simplified attention parameterizations: diagonal (Boix-Adsera

et al., 2023), position-only softmax (Jelassi et al., 2022), position-free softmax (Tian et al., 2023), component-wise (Marion et al., 2025), or linear (Zhang et al., 2024; Shen et al., 2025) attention. Closest to our setting, Zucchet et al. (2026) consider the same data model with position-only softmax and $h = 1$, characterizing escape from the uniform-attention initialization via a local Taylor approximation. The multi-head setting is closer to our focus but comparatively less developed. Chen et al. (2024b) and Zhang et al. (2025) study in-context linear regression with multi-head softmax and linear attention, respectively. Chen et al. (2024c) analyze the training dynamics of softmax attention on in-context $n$-gram tasks, and Varre & Flammarion (2025) study incremental learning on a related associative recall task.

**Discussion.** We study the saddle-to-saddle dynamics of multi-head softmax attention in a positional setting with an importance hierarchy over past positions. This setting reveals a competitive-to-cooperative transition: heads first compete for a dominant sparse pattern, then specialize sequentially via offshooting, with non-offshooting heads compensating through their value matrices. These structural signatures go beyond the mere presence of staged learning or head specialization. Capturing these signatures requires jointly analyzing value and attention parameter dynamics in a multi-head system; they would be obscured by freezing parameter blocks to impose stage-wise training or by reductions that decouple head trajectories, e.g., via initialization or parameterization. This competition is also distinct from circuit-level competition between algorithmic solutions (Reddy, 2024; Park et al., 2025): here, the competing units are symmetric heads sharing the same input and output domain. More broadly, our results give a controlled temporal example of the head-level interactions that Chakrabarti & Balachundar (2026) frame as a multi-player game.

## 5. Conclusion

In this work, we introduced a simple but rich task in which transformers need to implement multiple sparse attention patterns. We showed that it exhibits features of position-dependent incremental learning in transformers: learning begins in a competitive regime, where all heads focus on the most important pattern, and later becomes cooperative, with offshooting heads specializing in other patterns. To analyze this mechanism, we used a simplified architecture and regression objective, deriving surrogate differential equations that capture the same qualitative transition. Together, these results provide a controlled account of how sparse attention patterns and head specialization can emerge through training dynamics. Extending this analysis to more realistic architectures, objectives, and data distributions remains an important direction for future work.

## Acknowledgments

This work was partially funded by an unrestricted gift from Coefficient Giving, and the grant number 212111 from the Swiss National Science Foundation. Oğuz Kaan Yüksel is supported by the SwissAI Fellowship.

Due to an oversight during submission and the conference author-list policy, this proceedings version lists only the submitting author.

## Impact Statement

This paper presents work whose goal is to advance the field of Machine Learning. There are many potential societal consequences of our work, none of which we feel must be specifically highlighted here.

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

## Organization of the Appendix

The appendix is organized as follows.

- Section A provides the experimental details for the experiments in Sections 2.3 and 2.4.

- Section B presents additional experiments varying the data regime (infinite data, weight decay), the importance and interval structure (reverse order, non-uniform $\alpha$, overlapping intervals), the depth (two-block transformers), and the optimizer (SGD).

- Section C gives examples of the non-identifiability of the task.

- Section D simulates the regression flow, verifies the theoretical predictions against it, and compares the resulting trajectories with the minimal transformer.

- Appendix E and Section E.3 provide proofs of the theoretical results.

- Appendix F discusses how the initialization in our main theorems can be relaxed.

## A. Experimental Details

The full model has a standard single-layer transformer decoder architecture as discussed in Section 2.3. It uses absolute positional encodings with learnable embedding and unembedding matrices and has the configuration shown in Table 3. The minimal model, as described in Section 2.2, removes layer normalization, dropout, residual connections, key and output attention matrices, and the MLP layer. It uses one-hot positional encodings and does not have embedding and unembedding matrices. Both the full model and the minimal model are trained with the same optimization hyperparameters listed in Table 2, and the same synthetic data generation process described in Table 1. The main difference in the learning task between the two models is the interval lengths $|I(k)|$ of the Markov process: the full model uses intervals of length 4, while the minimal model uses intervals of length 2, as summarized in Table 4.

We train the $n$-gram models using the same architecture and optimization hyperparameters as the full transformer model, but train with windows of size $n$ sliding over the full sequence. The source code to reproduce our experiments is available at https://github.com/tml-epfl/sparse-attention-dynamics.

*Table 1.* Synthetic dataset parameters

| Parameter | Value |
|---|---|
| Heads $h$ | 3 |
| Dictionary size $d$ | 50 |
| Multiplicative constant $m$ | 1.7 |
| Base scale $b_0$ | 10 |
| Sequence length $T$ | 20 |
| Train samples | 9000 |
| Test samples | 3000 |
| Seed | 0 |

*Table 2.* Optimization hyperparameters

| Parameter | Value |
|---|---|
| Steps | 2000 |
| Batch size | 3000 |
| Gradient clipping | 1.0 |
| Optimizer | AdamW |
| Weight decay | 0.01 |
| Learning rate | 0.003 |
| Scheduler | ReduceLROnPlateau |
| Patience | 10 |
| Factor | 0.5 |

*Table 3.* Transformer configuration

| Parameter | Value |
|---|---|
| Hidden dimension | 255 |
| Feedforward dimension | 64 |
| Dropout | 0.1 |
| Initialization scale | 1 |
| Number of blocks | 1 |
| Number of heads | 3 |

*Table 4.* Markov process intervals

| | Full | Minimal |
|---|---|---|
| $w$ | 12 | 6 |
| $I(1)$ | $\{1, 2, 3, 4\}$ | $\{1, 2\}$ |
| $I(2)$ | $\{5, 6, 7, 8\}$ | $\{3, 4\}$ |
| $I(3)$ | $\{9, 10, 11, 12\}$ | $\{5, 6\}$ |

# B. Additional Experiments

We run additional experiments to study incremental learning behavior under different settings. In particular, we study the effect of infinite data versus finite data, different orders of importance with non-uniform interval lengths, and the impact of weight decay.

### B.1. Infinite Data

Instead of training on a finite dataset of 9000 samples, we train the model with infinite data by sampling a new batch of data at each step. This removes any effect of overfitting on incremental learning. We observe in Figure 6 and Figure 7 that the model still exhibits the same behavior. This experiment is run with the minimal architecture described in Section 2.4.

### B.2. Reverse Order

We reverse the order of importance of the intervals such that the most important interval is the furthest one. Figure 8 and Figure 9 show the results when $I(3) = \{12, 13\}$, $I(2) = \{8, 9, 10, 11\}$, and $I(1) = \{0, 1, 2, 3, 4, 5, 6, 7\}$, which reveal the same behavior as the original order. We also note that it is generally easier to observe incremental learning behavior when the most important interval is the furthest one. This indicates that the learning dynamics are impacted by the sequential structure of the task. This experiment is run with the full architecture described in Section 2.3.

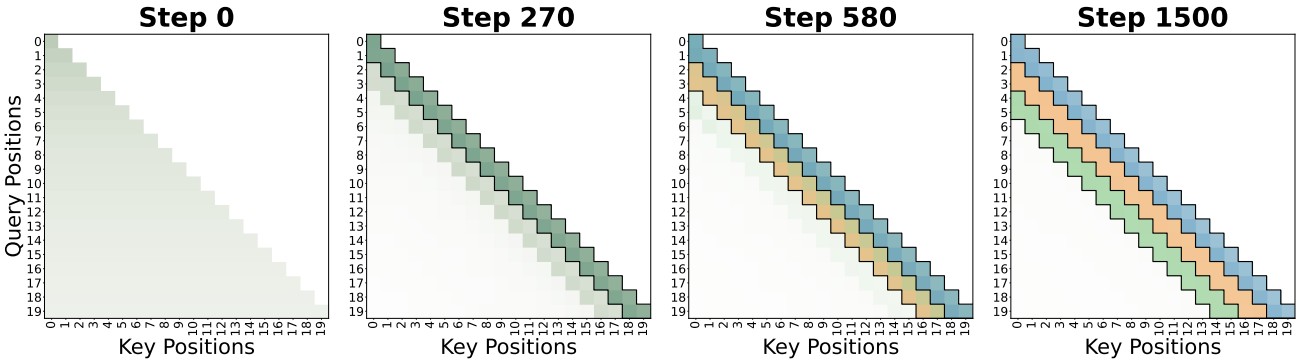

*Figure 6.* Attention patterns over the training steps with online sampling of data.

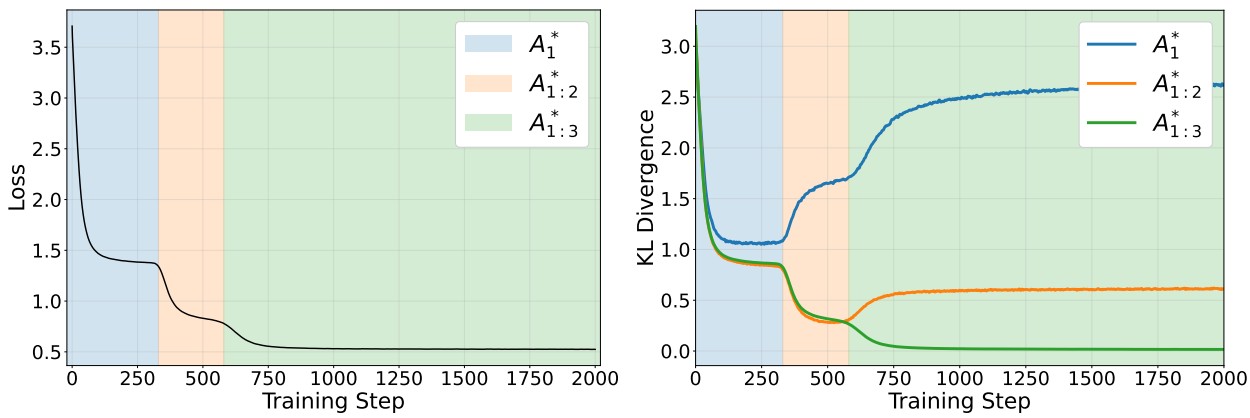

*Figure 7.* Validation loss and KL divergence over the training steps with online sampling of data.

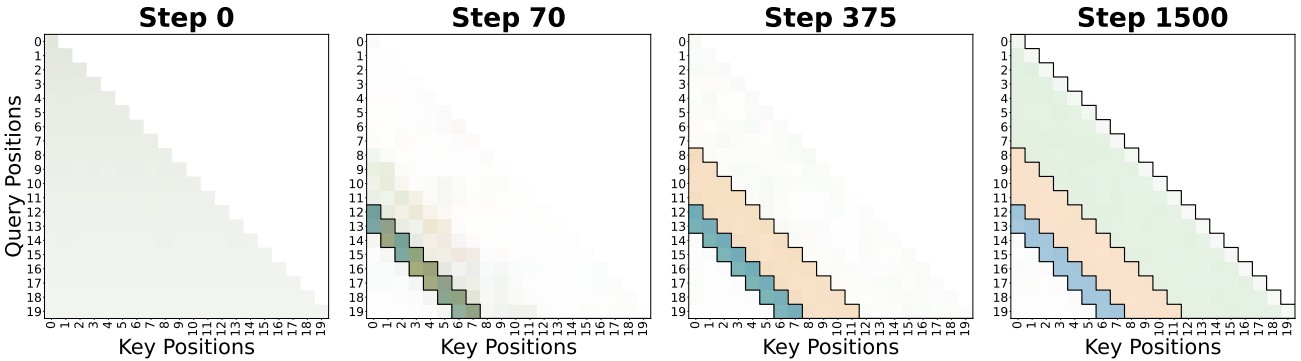

*Figure 8.* Attention patterns over the training steps with reversed order of importance and varying interval lengths.

### B.3. Weight Decay

We also study the impact of weight decay on the learning dynamics. We observe almost no difference in the learning dynamics when weight decay is not applied so we do not report the results.

### B.4. Two-block Transformers

We train 2-block minimal and full transformers with the same configuration as in Section A but adjusting the learning rate and number of training examples. Figures 10 and 11 show that the incremental learning behavior is similar to the 1-block case. We observe that the first region corresponding to the first feature matrix is less pronounced.

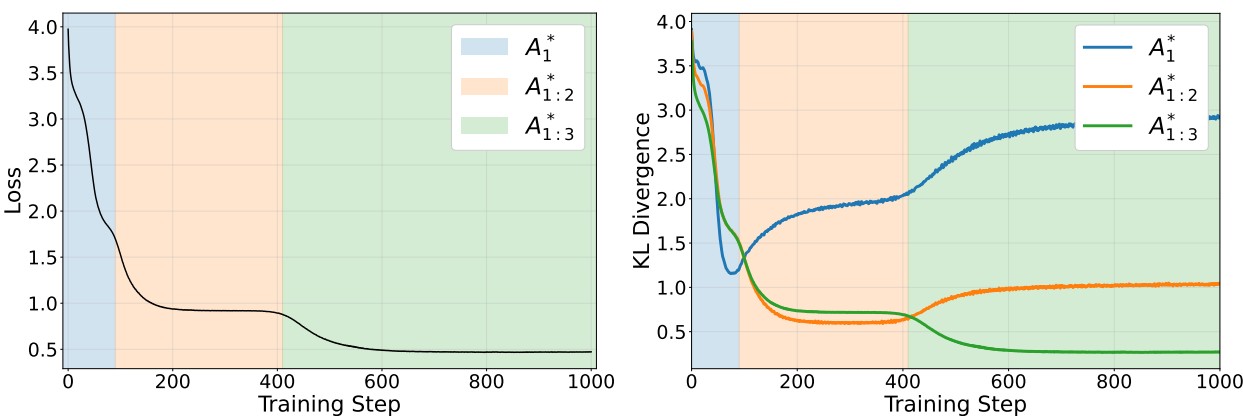

*Figure 9.* Validation loss and KL divergence over the training steps with reversed order of importance and varying interval lengths.

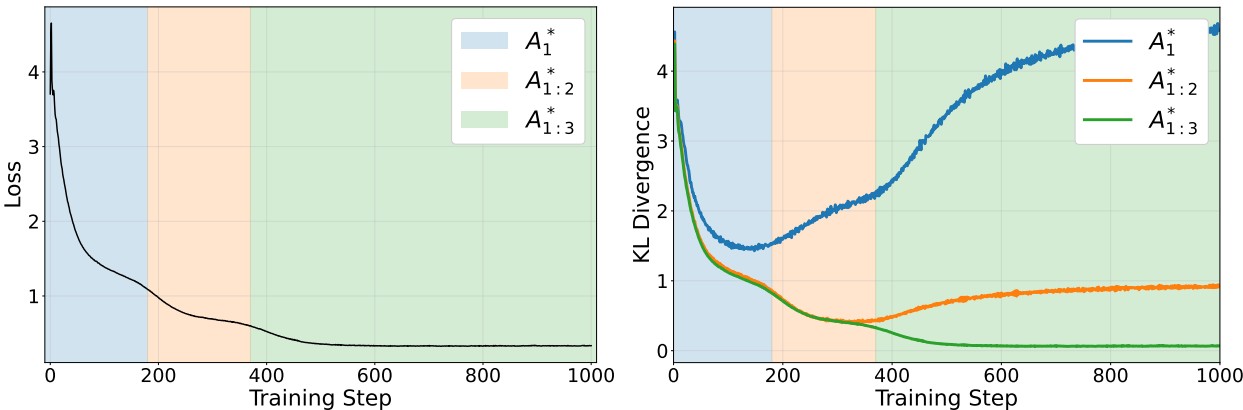

*Figure 10.* Validation loss and KL divergence over the training steps for a 2-layer minimal transformer.

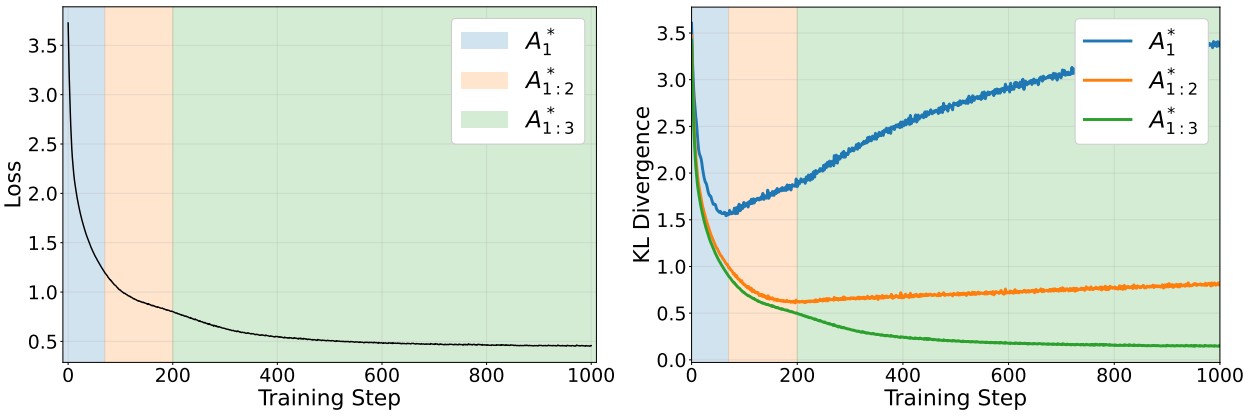

*Figure 11.* Validation loss and KL divergence over the training steps for a 2-layer full transformer.

### B.5. Non-uniform $\alpha$ values

We run experiments with $\alpha = [0.7, 0.3]$ in Figure 12 and observe that the model still exhibits incremental learning. In Figure 13, we observe the checkered pattern where heads focus more attention on the position with the highest $\alpha$ value.

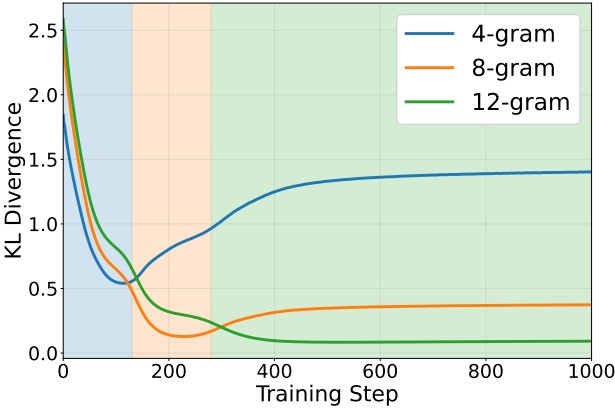

*Figure 12.* KL divergence over the training steps with non-uniform $\alpha$ values.

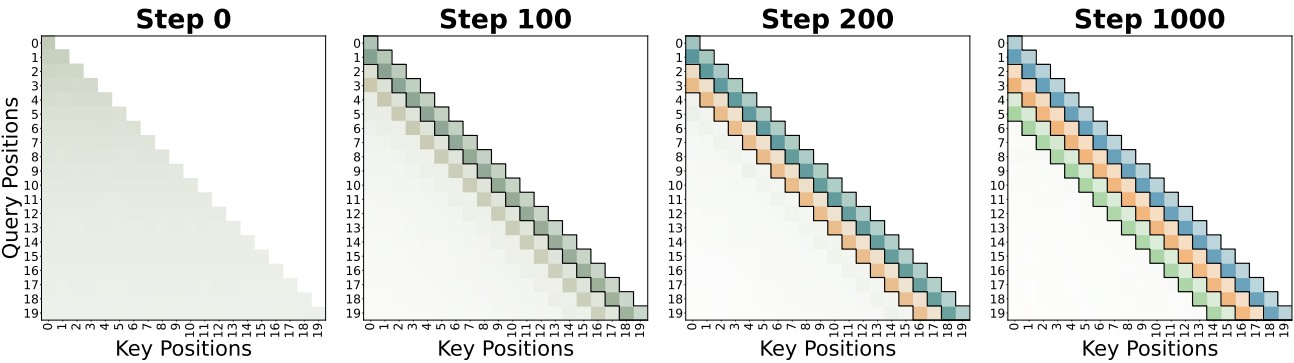

*Figure 13.* Attention patterns over the training steps with non-uniform $\alpha$ values.

## B.6. Overlapping Intervals

We run experiments with overlapping intervals where $I(1) = \{5, 6, 7, 8\}$, $I(2) = \{3, 4, 5, 6\}$, and $I(3) = \{1, 2, 3, 4\}$. This corresponds to interval lengths of 4 with an overlap or stride of 2. We try learning transformers with three or four heads. We observe in Figure 14 that the model with four heads still exhibits incremental learning behavior. Similar results are observed for the model with three heads and thus omitted. Attention patterns in Figures 15 and 16 reveal the different learning orderings for three and four heads. When the intervals are overlapping, it is unclear which positions are statistically the most significant, and transformers may follow different solutions based on the feature matrices.

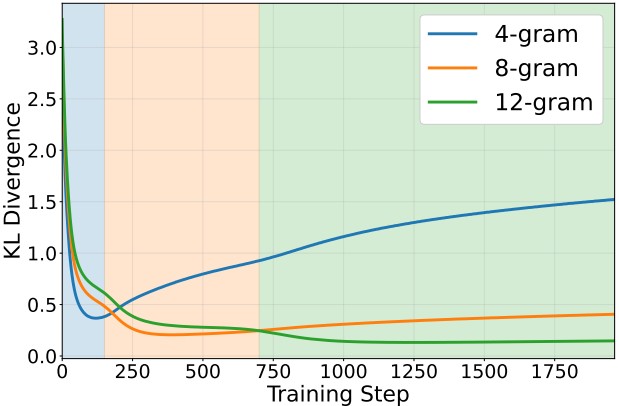

*Figure 14.* KL divergence over the training steps with intervals of size 4 and overlap of 2 for a transformer model with 4 heads.

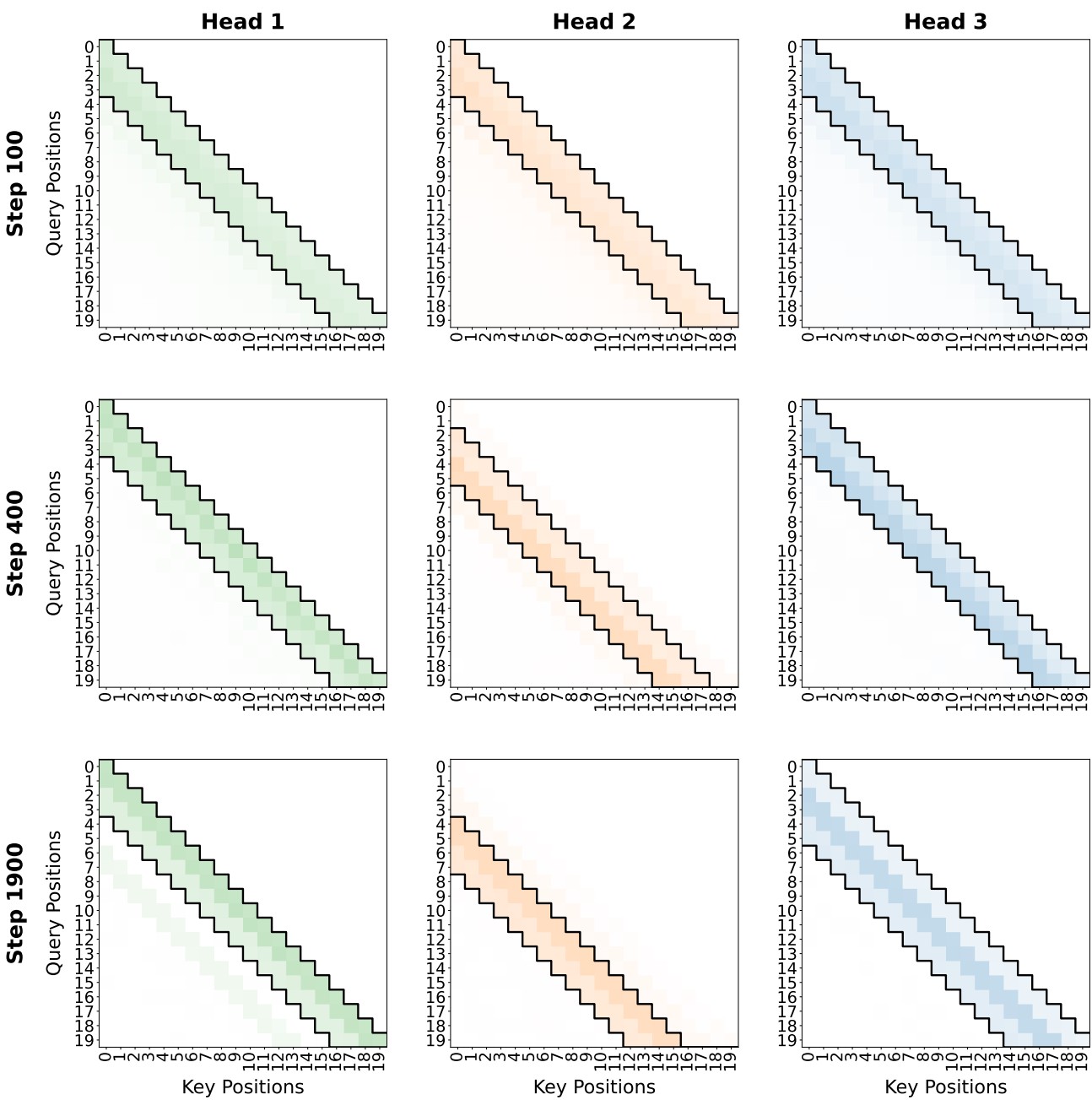

*Figure 15.* Attention patterns for 3 heads over the training steps with overlapping intervals.

## B.7. Stochastic Gradient Descent (SGD)

We run experiments with SGD optimizer instead of AdamW. We observe in Figure 17 and Figure 18 that the quantitative behavior of incremental learning is the same.

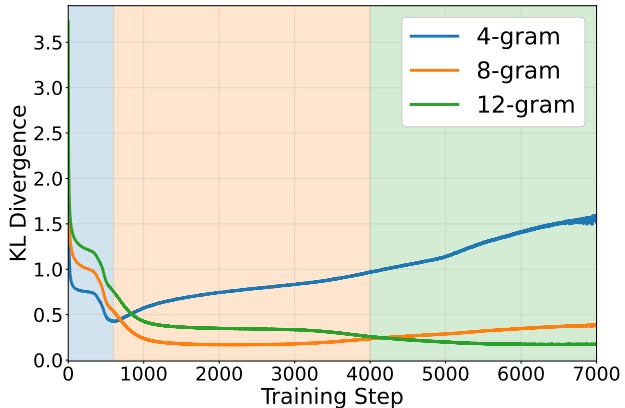

*Figure 16.* Attention patterns for 4 heads over the training steps with overlapping intervals.

*Figure 17.* KL divergence over the training steps with SGD optimizer.

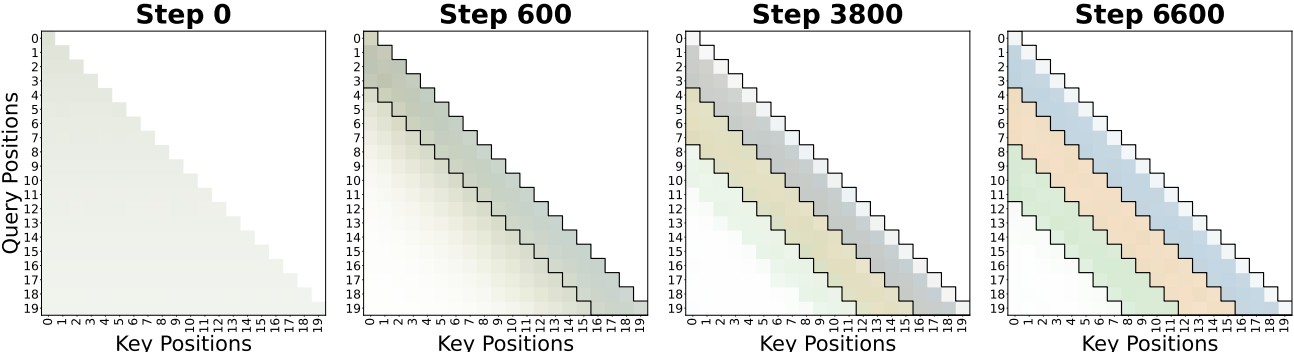

*Figure 18.* Attention patterns over the training steps with SGD optimizer.

## C. Identifiability

The data generation process defined in Equation (1) is not identifiable: different choices of partitions $I(k)$ and feature matrices $A_k^\star$ can induce the same conditional distribution. A trivial source of non-identifiability is permutation: jointly permuting the pairs $\{(I(k), A_k^\star)\}_{k=1}^h$ leaves the sum in Equation (1) unchanged.

Beyond permutations, there exist non-trivial decompositions in which the partitions $I(k)$ overlap and the attention weights $\alpha_i$ spread across multiple positions. Consider $h = 2$ with the sparse, disjoint decomposition

$$I(1) = \{1\}, \quad I(2) = \{2\}, \quad \alpha_1^{(1)} = \alpha_2^{(2)} = 1,$$

and feature matrices $(A_1^\star, A_2^\star)$. The resulting logits are

$$A_1^\star x_{t-1} + A_2^\star x_{t-2}.$$

Now take the alternative decomposition with overlapping partitions

$$I(1) = I(2) = \{1, 2\}, \quad \alpha^{(1)} = (2/3, 1/3), \quad \alpha^{(2)} = (1/3, 2/3),$$

and modified feature matrices

$$B_1^\star = 2A_1^\star - A_2^\star, \quad B_2^\star = -A_1^\star + 2A_2^\star.$$

The resulting logits are

$$B_1^\star \left( \tfrac{2}{3} x_{t-1} + \tfrac{1}{3} x_{t-2} \right) + B_2^\star \left( \tfrac{1}{3} x_{t-1} + \tfrac{2}{3} x_{t-2} \right) = A_1^\star x_{t-1} + A_2^\star x_{t-2},$$

matching the original distribution. In this alternative representation, each head attends to both positions, and the feature matrices are linear combinations of the original. More generally, any invertible row-stochastic matrix

$$\begin{pmatrix} a & 1-a \\ b & 1-b \end{pmatrix},$$

with $a \neq b$ yields an equivalent representation, and analogous constructions exist for $h > 2$. As shown in Section 2.3, training consistently recovers the sparse, disjoint decomposition despite these non-sparse representations.

## D. Simulations and Comparison with Transformers

In this section, we numerically simulate the regression gradient flow analyzed in Section 3.1, verify that our theoretical predictions match these simulations, and compare the resulting trajectories with those of the minimal transformer trained on the classification task from Section 2.3.

### D.1. Numerical Simulations of the Regression Flow

We present numerical simulations of the gradient flow dynamics of the loss in Equation (3) with the following parameters: $d = 50$, $T = 40$, $h = 3$, $|I(k)| = 1$ for all $k \in [h]$, and $m = 1.7$. We initialize the value parameters $V_i$ to 0 and the attention patterns $s_i$ to $\frac{1}{T} 1_T + \epsilon_i$ where $\epsilon_i$ are sampled from Gaussian distribution with zero-mean and $\epsilon I_T$ covariance with $\epsilon = 10^{-6}$. Figure 19 shows the evolution of the attention patterns $s_k$, the value parameters $V_k$, and the loss over time.

The results align with the transformer experiments in Section 2.3. Similar to the transformer experiments, the heads first learn from position (1), then position (2), and finally position (3). The time scales of these stages are clearly separated where the first stage is the fastest and the third stage is the slowest. Notably, at first, all heads try to learn from the position (1), as it is related to the most important feature. After this competitive phase, the heads start to learn from the position (2) and then the position (3) where they specialize in different patterns. Here, they cooperate to learn from the position (3). In particular, the first head offsets feature (3) as the third head's residual attention on the first position results in a cross term.

Figure 20 overlays the predictions of our simplified analysis on the full regression simulation. We pick $t = 25$ as the start of the second (cooperative) stage and $t = 300$ as the start of the third stage; at each boundary, we initialize the simplified coupled ODEs of Section 3.3 from the theoretical initialization for that stage and integrate them forward, plotting the resulting trajectories as dashed curves. The simplified dynamics quantitatively trace the full ODE across both attention positions $s_k$ and value parameters $V_k$, including the non-monotonic dips in non-offshooting heads during the cooperative phase. This indicates that our theoretical simplifications preserve the quantitative behavior of the underlying dynamics, not just the qualitative stage structure.

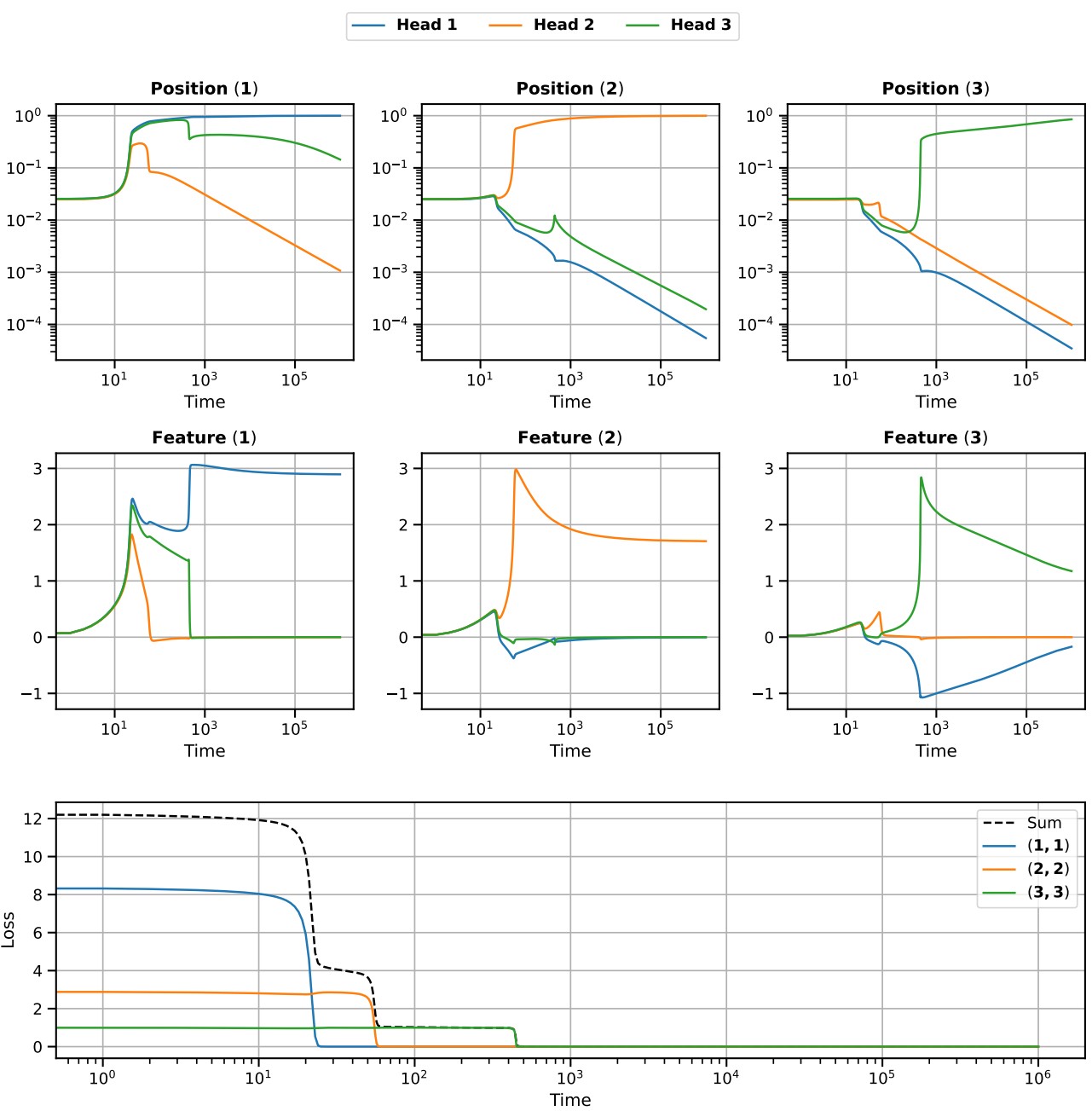

*Figure 19.* (Top) Per-head attention mass $s_k$ on positions $(1), (2), (3)$ over time. (Middle) Per-head value-matrix alignments $\langle V_k, V_j^\star \rangle$ with the three feature directions. (Bottom) The evolution of the loss over time. We only plot the relevant coordinates of $s_k$ and $V_k$ for clarity. We decompose the loss into the (feature, position) contributions which are plotted in the color of the heads that learn these contributions.

## D.2. Comparison with the Classification Transformer

We now compare the regression-flow simulation in Figure 19 with the corresponding trajectories of the minimal transformer trained on the classification task in Figure 21. Despite the differences in architecture, loss, and parameterization, the two settings agree on the structural signatures of the dynamics rather than only on the existence of stages. First, both exhibit a clear competitive phase in which all three heads concentrate on the same most important positions before any specialization occurs, with attention to the remaining positions decaying simultaneously across heads. Second, the offshooting order is the

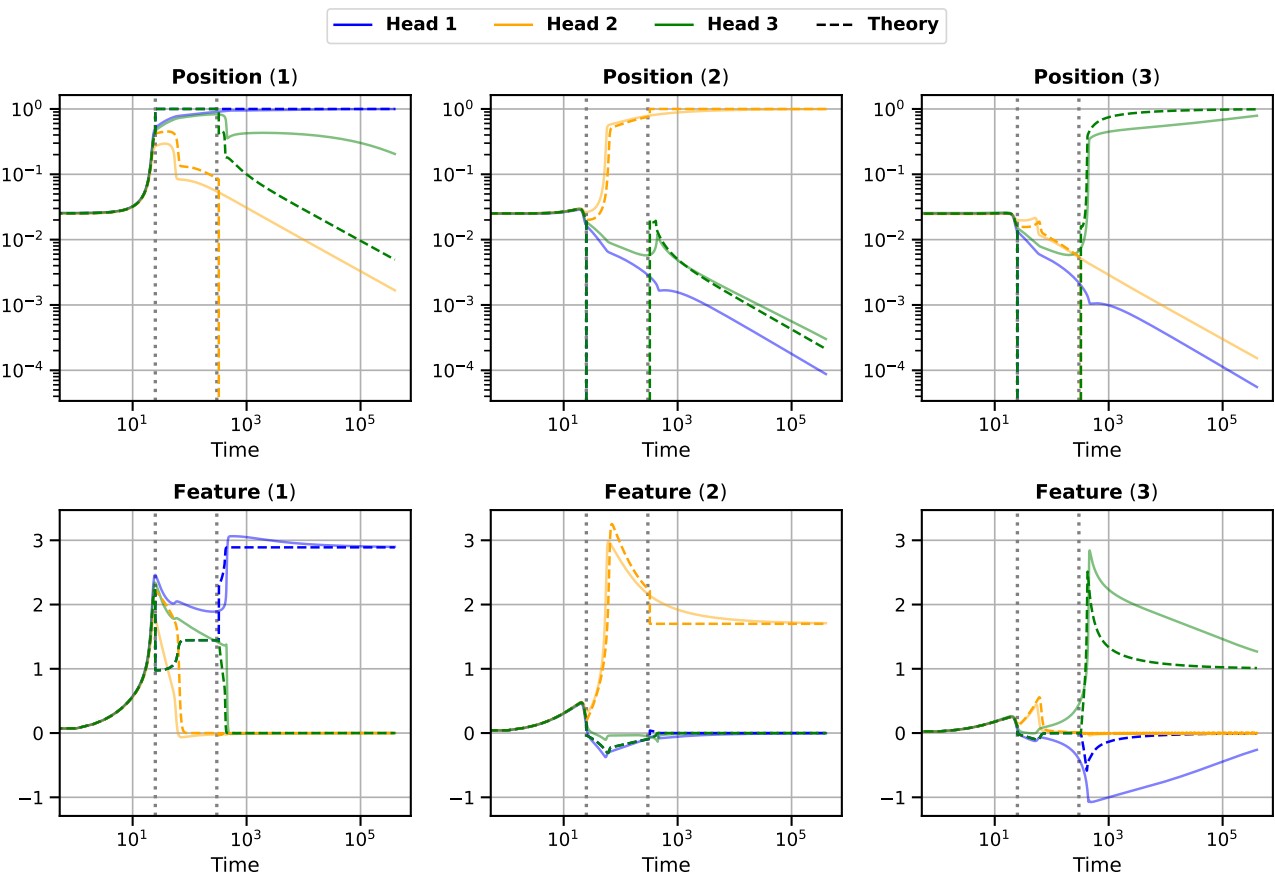

*Figure 20.* Quantitative comparison between the regression gradient-flow simulation (solid) and the simplified theoretical dynamics (dashed). At each stage boundary $t = 25$ for the second stage and $t = 300$ for the third, we initialize the simplified coupled ODEs of Section 3.3 from the theoretical initialization for that stage and integrate them forward; the dashed curves are the resulting trajectories. (Top) Per-head attention mass $s_k$ on positions (1), (2), (3) over time. (Bottom) Per-head value-matrix alignments $\langle V_k, V_j^\star \rangle$ with the three feature directions. Vertical dotted lines mark the stage boundaries. The theoretical trajectories quantitatively track the full ODE, capturing both convergence to each saddle and the compensatory dips in non-offshooting heads during the cooperative phase.

same in both: heads leave the shared pattern one at a time, in the order dictated by the feature importances $m_1^\star > m_2^\star > m_3^\star$, with the second-most important position acquired before the least important one. Third, the value matrices reproduce the cooperative compensation predicted by the theory: when the offshooting head locks onto a new feature, the non-offshooting heads reduce their alignment with that same feature direction to cancel the cross-term induced by the offshooting head's residual attention. This is visible in Figure 19 as the negative dips in $V_k$ for the heads that did not offshoot in the current stage, and in Figure 21 as the corresponding dips in the value-matrix alignments of Heads 1 and 3 during Stage 2 and of Head 3 during Stage 3. The comparison is qualitative: the regression flow uses a continuous-time gradient flow on a population objective with singleton groups, while the transformer is trained on a finite sample with cross-entropy and absolute positional encodings, so axis scales and timescales differ by construction.

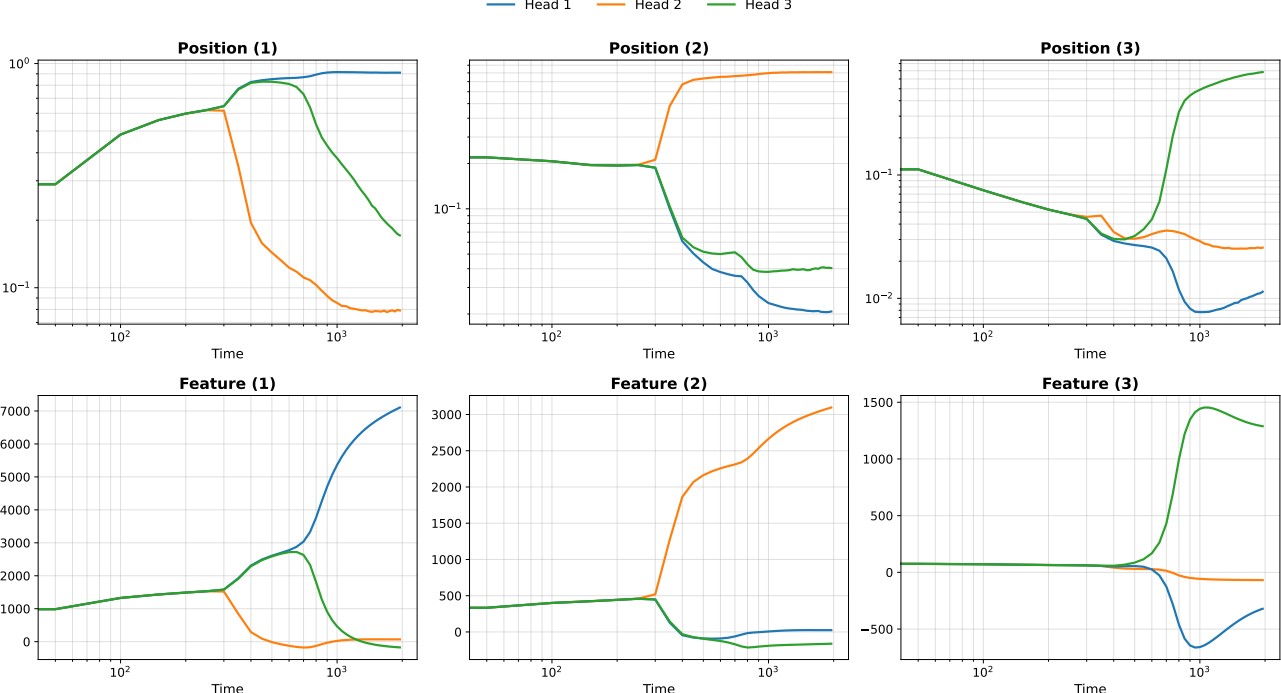

*Figure 21.* Same quantities as in Figure 19, but measured on the minimal model with a small initialization on the classification task rather than on the regression gradient flow. (Top) Per-head attention mass on positions in $I(1), I(2), I(3)$ over training. (Bottom) Per-head alignment of the value matrix with the feature directions $A_1^\star, A_2^\star, A_3^\star$. The transformer reproduces the structural signatures predicted by the theory: (i) a competitive phase in which all heads concentrate on $I(1)$, the most important positions; (ii) a sequential offshoot, with Head 2 acquiring $I(2)$ before Head 3 acquires $I(3)$; (iii) compensatory decreases in the non-offshooting heads' value alignments (e.g., Heads 1 and 3 reduce their alignment with $A_2^\star$ as Head 2 grows, and Head 3 reduces its alignment with $A_1^\star$ as Head 1 grows), matching the cooperative mechanism analyzed in Section 3.3. The comparison with Figure 19 is qualitative: the two settings differ in architecture, loss, and parameterization, so axis scales and timescales are not aligned.

# E. Deferred Proofs

We start with the proof of Proposition 1 and some elementary results on the operation $\Pi$. Recall that we assume $s_i^\star$ are one-hot in Section 3.1. That is, in the sequel, $\|s_i^\star\|^2 = 1$.

**Proposition 1.** *The gradient flow dynamics of the loss in Equation* (3) *is equivalent to those of* $\mathcal{L}(\theta) = \dfrac{1}{2}\|\boldsymbol{G} - \boldsymbol{P}\|_F^2$

$$\text{where } \boldsymbol{P} = \sum_{k=1}^{h} V_k \otimes s_k, \text{ and } \boldsymbol{G} = \sum_{k=1}^{h} m_k^\star \left(V_k^\star \otimes s_k^\star\right).$$

*Proof.* We start by some computations. Note that for any vectors $v_1, v_2 \in \mathbb{R}^T$, we have:

$$\mathbb{E}\left[(Xv_1)(Xv_2)^\top\right] = \sum_{i=1}^{T}\sum_{j=1}^{T}(v_1)_i(v_2)_j\mathbb{E}\left[x_i x_j^\top\right]$$
$$= \langle v_1, v_2 \rangle I_d.$$

Also, for any vectors $v_1, v_2 \in \mathbb{R}^T$ and any matrix $Q \in \mathbb{R}^{d \times d}$, we have:

$$\mathbb{E}\left[v_1^\top X^\top Q X v_2\right] = \sum_{i=1}^{T}\sum_{j=1}^{T}(v_1)_i(v_2)_j \mathbb{E}\left[x_i^\top Q x_j\right]$$

$$= \sum_{i=1}^{T}\sum_{j=1}^{T}(v_1)_i(v_2)_j \operatorname{Tr}\left(Q\mathbb{E}\left[x_j x_i^\top\right]\right)$$

$$= \langle v_1, v_2\rangle \operatorname{Tr}(Q) .$$

By selecting $v_2 = e_i$ for all $i \in [d]$, we get:

$$\mathbb{E}\left[v_1 X^\top Q X\right] = \operatorname{Tr}(Q)v_1 .$$

First, the derivative with respect to $V_i$ is as follows:

$$\frac{\partial \mathcal{L}(\theta)}{\partial V_i} = \mathbb{E}_{X,\xi}\left[\left(f_\theta(X) - f^\star(X,\xi)\right)\left(X s_i\right)^\top\right]$$

$$= \sum_{j=1}^{h}\langle s_i, s_j\rangle V_j - \sum_{j=1}^{h}m_j^\star\langle s_i, s_j^\star\rangle V_j^\star .$$

Next, the derivative with respect to $q_i$ is as follows:

$$\frac{\partial \mathcal{L}(\theta)}{\partial q_i} = \left(\operatorname{diag}(s_i) - s_i s_i^\top\right)\mathbb{E}_{X,\xi}\left[X^\top V_i^\top\left(f_\theta(X) - f^\star(X,\xi)\right)\right]$$

$$= \left(\operatorname{diag}(s_i) - s_i s_i^\top\right)\left(\sum_{j=1}^{h}\langle V_i, V_j\rangle s_j - \sum_{j=1}^{h}m_j^\star\langle V_i, V_j^\star\rangle s_j^\star\right) .$$

Then, the gradient flow dynamics are as follows:

$$\dot{V}_i = -\nabla_{V_i}\mathcal{L}(\theta) = \left(\boldsymbol{G} - \boldsymbol{P}\right) s_i$$

$$\dot{q}_i = -\nabla_{q_i}\mathcal{L}(\theta) = \Pi(s_i)\left(V_i^\top\left(\boldsymbol{G} - \boldsymbol{P}\right)\right) .$$

This can be seen as a gradient descent flow on the following loss:

$$\mathcal{L}(\theta) = \frac{1}{2}\|\boldsymbol{G} - \boldsymbol{P}\|_F^2 .$$

$\square$

**Lemma 2.** *Let $s$ be a vector with non-negative entries and $\|s\|_1 = 1$. Then, the kernel space of $\Pi(s) = \operatorname{diag}(s) - s s^\top$ is*

$$\ker\left(\Pi(s)\right) = \operatorname{span}\left(\{e_j : \langle e_j, s\rangle = 0\}\right) \cup \operatorname{span}\left(\sum_{j:\langle e_j, s\rangle > 0} e_j\right) .$$

*Furthermore, if $\|s\|_1 < 1$,*

$$\ker\left(\Pi(s)\right) = \operatorname{span}\left(\{e_j : \langle e_j, s\rangle = 0\}\right) .$$

*Proof.* The proof follows trivially from a rank analysis. $\square$

**Lemma 3.** *Let $s$ be a vector on the simplex that satisfies $s_i \geq s_j$ for all $j \in [h]$. Then, for any vector $v$ that satisfies $v_i \geq v_j$ for all $j \in [h]$, we have for all $j \in [h]$:*

$$\left(\Pi(s)v\right)_i \geq \left(\Pi(s)v\right)_j .$$

*Proof.* We have the following computations:

$$(\Pi(s)v)_i = s_i\,(v_i - \langle s, v\rangle)$$
$$(\Pi(s)v)_j = s_j\,(v_j - \langle s, v\rangle)\,.$$

Then, we have:

$$(\Pi(s)v)_i - (\Pi(s)v)_j \geq (s_i - s_j)\,(v_i - \langle s, v\rangle) \geq 0\,.$$

$\square$

### E.1. Boundedness

In this section, we prove Theorems 2 and 3, which are required to establish boundedness of the dynamics.

**Theorem 2.** *Assume that the following holds for $\epsilon \ll 1$:*

$$\forall k \in [h] : \|V(0) - V_k(0)\|_F \leq \epsilon \ \text{ and } \ \|s(0) - s_k(0)\|_2 \leq \epsilon\,.$$

*Then, there exists a constant $c_1$ such that $\forall t \in \left[0, \frac{1}{-c_1 \log \epsilon}\right]$:*

$$\|V_k(t) - V(t)\|_F \leq \epsilon e^{c_1 t} \quad \text{and} \quad \|s_k(t) - s(t)\|_2 \leq \epsilon e^{c_1 t}\,.$$

*Proof.* We write the flow of $V_i$ and $s_i$ in terms of the flow of $V$ and $s$ using new variables:

$$W_i = V_i - V\,, \quad z_i = s_i - s\,.$$

Let $\epsilon$ be the following quantity:

$$\epsilon = \max_{j\in[h]} \max\{\|W_j\|_F, \|z_j\|\}\,.$$

We are interested in the regime where $\epsilon \ll 1$.

Recall that $\phi(V, s)$ defined in Equation (11) is always non-decreasing. Therefore, $V$ cannot grow larger than $\dfrac{\boldsymbol{G}s}{h\|s\|^2}$ in norm or otherwise $\phi(V, s)$ would decrease. This is the optimal value of $V$ for a particular $s$. Thus, we have a time-independent upper bound $|V| \leq \max_s \dfrac{\boldsymbol{G}s}{h\|s\|^2} = \dfrac{m_1^\star}{h}$.

Then, the flow of $W_i$ and $z_i$ is as follows:

$$\dot{W}_i = \boldsymbol{G}z_i - \boldsymbol{P}s_i + h\|s\|^2 V\,,$$
$$\dot{z}_i = \Pi(s_i)^2 \left(V_i^\top (\boldsymbol{G} - \boldsymbol{P})\right) - \Pi(s)^2 \left(V^\top \boldsymbol{G} - h\|V\|^2 s\right)\,.$$

Note that $\boldsymbol{P}$ can be rewritten as follows:

$$\boldsymbol{P} = \sum_{j=1}^h V_j \otimes s_j = hV \otimes s + \left(\sum_{j=1}^h W_j\right) \otimes s + V \otimes \left(\sum_{j=1}^h z_j\right) + \left(\sum_{j=1}^h W_j \otimes z_j\right)\,.$$

This implies that:

$$V^\top \boldsymbol{P} = h\|V\|^2 s + \mathcal{O}\left(\epsilon + \epsilon^2\right)\,, \quad \boldsymbol{P}s = h\|s\|^2 V + \mathcal{O}\left(\epsilon + \epsilon^2\right)\,.$$

We can rewrite the flow of $z_i$ as follows:

$$\dot{z}_i = \left(\Pi(s_i)^2 - \Pi(s)^2\right)\left(V_i^\top (\boldsymbol{G} - \boldsymbol{P})\right) + \Pi(s)^2 \left(W_i^\top \boldsymbol{G} - V_i^\top \boldsymbol{P} + h\|V\|^2 s\right)\,.$$

Therefore, we have:

$$\dot{W}_i = \mathcal{O}(\epsilon)\,, \quad \dot{z}_i = \mathcal{O}(\epsilon)\,.$$

The norms of $W_i$ and $z_i$ then evolve as follows:

$$\widehat{\|\dot{W}_i\|} = \frac{\dot{W}_i^\top W_i}{\|W_i\|} \leq \|\dot{W}_i\| = \mathcal{O}(\epsilon) \,.$$

We similarly derive that $\|\dot{z}_i\| = \mathcal{O}(\epsilon)$.

This implies that $\epsilon$ satisfies the equation:

$$\dot{\epsilon} \leq C\epsilon \,, \quad \text{as long as } \epsilon \ll 1 \,,$$

where $C$ is a constant that depends on the problem parameters $h$ and $\mathbf{G}$. By Grönwall's inequality, we have:

$$\epsilon(t) \leq \epsilon(0)e^{Ct} \,, \quad \text{as long as } t \in \left[0, \frac{1}{-C\log \epsilon(0)}\right] \,.$$

$\square$

**Theorem 3.** *Assume that the following holds for $\epsilon \ll 1$:*

$$\forall k \in [h-1] : \|V(0) - V_k(0)\|_F \leq \epsilon, \|s_1^\star - s_k(0)\|_2 \leq \epsilon \,,$$
$$\text{and} \quad \|V'(0) - V_h(0)\|_F \leq \epsilon \,, \|s'(0) - s_h(0)\|_2 \leq \epsilon \,.$$

*Let $\Delta(t)$ be the deviation from the cooperative system in Equation (8):*

$$\Delta(t) = \max \left\{ \max_{k \in [h-1]} \{\|V_k(t) - V(t)\|_F, \|s_k(t) - s(t)\|_2\} \,, \right.$$
$$\left. \|V_h(t) - V'(t)\|_F, \|s_h(t) - s'(t)\|_2 \right\} \,.$$

*Assuming that $\|s'(t) - s_1^\star\| \geq \delta$ for all $t \in \mathbb{R}$, there exists a universal constant $c_1$ such that:*

$$\Delta(t) \leq \epsilon e^{c_1 t} \,, \quad \forall t \in \left[0, \frac{1}{-c_1 \log \epsilon}\right] \,.$$

*Proof.* We follow the same strategy as in the proof of Theorem 2. The new Lyapunov function is as follows:

$$\phi(V, V', s') = (h-1)m_1^\star \langle V, V_1^\star \rangle - \frac{(h-1)^2}{2}\|V\|_F^2$$
$$- (h-1)\langle s_1^\star, s' \rangle \langle V, V' \rangle + \langle V', \mathbf{G}s' \rangle - \frac{1}{2}\|s'\|^2\|V'\|_F^2 \,.$$

We have the following derivatives:

$$\nabla_V \phi(V, V', s') = (h-1)\dot{V} \,,$$
$$\nabla_{V'} \phi(V, V', s') = \dot{V'} \,,$$
$$\nabla_{s'} \phi(V, V', s') = V^\top \mathbf{G} - (h-1)\langle V, V' \rangle s_1^\star - \|V'\|^2 s' \,.$$

By a similar argument, we have that

$$\dot{\phi} = (h-1)\|\dot{V}\|^2 + \|\dot{V'}\|^2 + \|\Pi(s')\dot{s'}\|^2 \geq 0 \,.$$

This indicates that $\phi$ is non-decreasing. By a similar argument to Theorem 2, we establish an upper bound to $\phi$ and consequently the boundedness of the flow. Then, it is possible to show the noise process grows as $\mathcal{O}(\epsilon)$ where $\epsilon$ is the same quantity as in Theorem 2. $\square$

### E.2. Competitive Phase

In this section, we prove the main result of Section 3.2.

**Theorem 1.** *Assume that the initialization satisfies the following for all $k \in [h]$:*

$$\langle V(0), V_1^\star \rangle \geq \langle V(0), V_k^\star \rangle, \quad \langle s(0), s_1^\star \rangle \geq \langle s(0), s_k^\star \rangle. \tag{5}$$

*Then, the dynamics of $V$ and $s$ converge to the following fixed point:*

$$V(\infty) = \frac{m_1^\star}{h} V_1^\star, \quad s(\infty) = s_1^\star. \tag{6}$$

*Proof.* Let $\mathcal{R}$ be the following set:

$$\mathcal{R} = \{(V, s) \mid \forall k \in [h], \langle V, V_1^\star - V_k^\star \rangle \geq 0, \langle s, s_1^\star - s_k^\star \rangle \geq 0\}.$$

We prove that the flow is forward-invariant on $\mathcal{R}$.

Fix any $j \in [h]$. Let $w_j = \langle V, V_1^\star - V_j^\star \rangle$, $z_j = \langle s, s_1^\star - s_j^\star \rangle$, $r_j = \langle s \odot s, s_1^\star - s_j^\star \rangle$, $t_j = \langle s \odot s \odot s, s_1^\star - s_j^\star \rangle$. The flow of $w_j$ and $z_j$ are as follows:

$$\dot{w}_j = m_1^\star \langle s, s_1^\star \rangle - m_j^\star \langle s, s_j^\star \rangle - h\|s\|^2 w_j,$$
$$\dot{z}_j = (s_1^\star - s_j^\star)^\top \Pi(s)^2 \left( V^\top \boldsymbol{G} - h\|V\|_F^2 s \right).$$

Rewriting the derivative of $\dot{z}_j$:

$$\dot{z}_j = \left((s_1^\star - s_j^\star)^\top \operatorname{diag}(s) - z_j s^\top \right) \Pi(s) \left( V^\top \boldsymbol{G} - h\|V\|_F^2 s \right)$$
$$= (s_1^\star - s_j^\star)^\top \operatorname{diag}(s)^2 \left( V^\top \boldsymbol{G} - h\|V\|_F^2 s \right) - z_j s^\top \operatorname{diag}(s) \left( V^\top \boldsymbol{G} - h\|V\|_F^2 s \right)$$
$$+ \left(\|s\|^2 z_j - r_j\right) \left( V^\top \boldsymbol{G} s - h\|V\|_F^2 \|s\|^2 \right)$$
$$= m_1^\star \langle s_1^\star, s \rangle^2 \|s_1^\star\|^2 \langle V, V_1^\star \rangle - m_j^\star \langle s_j^\star, s \rangle^2 \|s_j^\star\|^2 \langle V, V_j^\star \rangle - h\|V\|_F^2 t_j$$
$$- z_j s^\top \operatorname{diag}(s) \left( V^\top \boldsymbol{G} - h\|V\|_F^2 s \right) + \left(\|s\|^2 z_j - r_j\right) \left( V^\top \boldsymbol{G} s - h\|V\|_F^2 \|s\|^2 \right).$$

On the boundary of $\mathcal{R}$, we have $w_j = 0$ or $z_j = 0$. If $w_j = 0$, then $\dot{w}_j \geq 0$ and if $z_j = 0$, then $r_j = t_j = 0$ and $\dot{z}_j \geq 0$. Therefore, a flow that has started in $\mathcal{R}$ will remain in $\mathcal{R}$ for all time.

Now, consider the following Lyapunov function:

$$\phi(V, s) = \langle V, \boldsymbol{G} s \rangle - \frac{h}{2}\|V\|_F^2 \|s\|^2. \tag{11}$$

The derivative of $\phi(V, s)$ is as follows:

$$\nabla_V \phi(V, s) = \boldsymbol{G} s - h\|s\|^2 V,$$
$$\nabla_s \phi(V, s) = V^\top \boldsymbol{G} - h\|V\|_F^2 s.$$

Therefore, the time derivative of $\phi$:

$$\dot{\phi}(V, s) = \|\dot{V}\|^2 + \|\Pi(s)\nabla_s \phi(V, s)\|^2 \geq 0.$$

$\phi$ is optimized when $V = \dfrac{\boldsymbol{G} s}{h\|s\|^2}$ which leads to a finite value upper bound on $\phi(V, s)$. Therefore, $\lim_{t \to \infty} \phi(V(t), s(t))$ is finite and the flow converges to a stationary point of $\phi$. That is, the flow converges to a point $(V_\infty, s_\infty)$ that satisfies:

$$\boldsymbol{G} s_\infty - h\|s_\infty\|^2 V_\infty = 0, \quad V_\infty^\top \boldsymbol{G} - h\|V_\infty\|_F^2 s_\infty \in \ker(\Pi(s_\infty)).$$

Note that we have the following equality:

$$(\boldsymbol{G}s_\infty)^\top \boldsymbol{G} = \sum_{j=1}^{h} m_j^\star \left\langle V_j^\star, \sum_{k=1}^{h} m_k^\star V_k^\star \langle s_k^\star, s_\infty \rangle \right\rangle s_j^\star = \sum_{j=1}^{h} (m_j^\star)^2 \langle s_j^\star, s_\infty \rangle s_j^\star \,.$$

Then, the stationary point $(V_\infty, s_\infty)$ satisfies

$$\sum_{j=1}^{h} (m_j^\star)^2 \langle s_j^\star, s_\infty \rangle s_j^\star - h^2 \|s_\infty\|^2 \|V_\infty\|_F^2 \langle s_j^\star, s_\infty \rangle s_j^\star \in \ker(\Pi(s_\infty)) \,. \tag{12}$$

We have proven that $\langle s_1^\star, s_\infty \rangle > 0$ as $\langle s_1^\star, s_\infty \rangle = \max_{k \in [h]} \langle s_k^\star, s_\infty \rangle$. From Lemma 2, $s_1^\star \notin \ker(\Pi(s_\infty))$ as there is at least one index $m \in [T]$ such that $\langle e_m, s_\infty \rangle > 0$ and $\langle e_m, s_1^\star \rangle > 0$. By projecting to the direction $s_1^\star$, Equation (12) implies

$$(m_1^\star)^2 \langle s_1^\star, s_\infty \rangle - h^2 \|s_\infty\|^2 \|V_\infty\|^2 \langle s_1^\star, s_\infty \rangle = 0 \,.$$

However, note that

$$h^2 \|s_\infty\|^2 \|V_\infty\|_F^2 = \frac{\|\boldsymbol{G}s_\infty\|^2}{\|s_\infty\|^2} \le \max_{\|s\|=1} \|\boldsymbol{G}s\|^2 = (m_1^\star)^2 \,,$$

with equality if and only if $s_\infty = s_1^\star$. Therefore, the flow converges to the stationary point

$$s = s_1^\star \,, \quad V = \frac{m_1^\star}{h} V_1^\star \,.$$

$\square$

### E.3. Cooperation Phase

In this section, we prove the remaining results in Section 3.3. First, we show convergence of the second head starting with the system in Equation (8). We then extend the analysis to arbitrary phases of the dynamics.

#### E.3.1. CONVERGENCE OF THE SECOND HEAD

Following Equation (7), we consider the following initialization scheme:

$$V(0) = \frac{1}{h-1} \left( m_1^\star V_1^\star - \langle s_1^\star, s'(0) \rangle V'(0) \right) \,, \quad V'(0) \approx V(0) \,, \quad s'(0) \approx s_1^\star \,. \tag{13}$$

Here, we note that $\dot{V}(0) = 0$. That is, $V(0)$ is at its optimal value given $V'(0)$ and $s'(0)$. The following lemma shows that $V$ stays close to its optimum through the trajectory:

**Lemma 1.** *Let* $\Delta(t) = V(t) - V^\star(t)$ *where*

$$V^\star(t) = \frac{1}{h-1} \left( m_1^\star V_1^\star - \langle s_1^\star, s'(t) \rangle V'(t) \right) \,.$$

*Assuming that* $\|s'(t) - s_1^\star\| \ge \delta$ *for all* $t \in \mathbb{R}$, *there exist constants* $c_1(\delta), c_2$ *such that*

$$\|\Delta(t)\|_F \le e^{-c_2 t} \|\Delta(0)\|_F + \frac{c_1(\delta)}{c_2} \,.$$

*Proof.* We compute the derivative of $\Delta$:

$$\dot{\Delta} = -(h-1)\|s_1^\star\|^2 \Delta + \frac{1}{h-1} \langle s_1^\star, s' \rangle \dot{V}' + \frac{1}{h-1} \langle s_1^\star, \dot{s}' \rangle V' \,.$$

Then, setting $c_2 = \frac{(h-1)}{2} \|s_1^\star\|^2$ and $c(t) = \frac{1}{h-1} \langle s_1^\star, s'(t) \rangle V'(t)$

$$\widehat{\|\Delta(t)\|_F^2} = 2\langle \dot{\Delta}(t), \Delta(t) \rangle = -2c_2 \|\Delta(t)\|_F^2 + 2\langle \dot{c}(t), \Delta(t) \rangle \,.$$

We bound the last term as follows:

$$\langle \dot{c}(t), \Delta(t)\rangle \le \|\dot{c}(t)\|_F \|\Delta(t)\|_F \,.$$

However, $\dot{c}(t)_F$ is uniformly bounded as in Theorem 3, so we get:

$$\widehat{\|\dot{\Delta}(t)\|_F^2} \le -2c_2\|\Delta(t)\|_F^2 + 2c_1\|\Delta(t)\|_F \,.$$

Set $u(t) = \|\Delta(t)\|_F - \dfrac{c_1}{c_2}$ and rewrite the inequality:

$$\dot{u}(t) \le -c_2 u(t) \,.$$

By Grönwall's inequality, we have the desired result. $\qquad\square$

Based on Lemma 1 and evidence from our numerical simulations, we approximate the full dynamics by a two-scale analysis where $V$ is optimized faster than $V'$ and $s'$, leading to Equation (9). Expanding $\Pi(s')$, we get

$$\Pi(s') = \Pi(s'_{(1)}) + \langle s_1^\star, s'\rangle \left( s_1^\star (s_1^\star)^\top - s_1^\star s_{(1)}'^\top - s'_{(1)} (s_1^\star)^\top \right) \,.$$

Since the $V_{(1)}'^\top G_{(1)} - \|V'\|_F^2 s'_{(1)}$ is perpendicular to the direction $s_1^\star$, we obtain:

$$\dot{s}' = \Pi(s') \left( \Pi(s'_{(1)}) - \langle s_1^\star, s'\rangle s_1^\star s_{(1)}'^\top \right) \left( V_{(1)}'^\top G_{(1)} - \|V'\|_F^2 s'_{(1)} \right) \,.$$

Writing out the update along the direction of $s_1^\star$:

$$
\begin{aligned}
\langle s_1^\star, \dot{s}'\rangle &= \langle s_1^\star, s'\rangle \|s_1^\star\|^2 \left( s_1^\star - s'_{(1)}\right)^\top \left( \Pi(s'_{(1)}) - \langle s_1^\star, s'\rangle s_1^\star s_{(1)}'^\top \right) \left( V_{(1)}'^\top G_{(1)} - \|V'\|_F^2 s'_{(1)} \right)\\
&= -\langle s_1^\star, s'\rangle \|s_1^\star\|^2 s_{(1)}'^\top \left( \Pi(s'_{(1)}) + \langle s_1^\star, s'\rangle \|s_1^\star\|^2 I \right) \left( V_{(1)}'^\top G_{(1)} - \|V'\|_F^2 s'_{(1)} \right) \,.
\end{aligned}
$$

The rest of the update follows:

$$
\begin{aligned}
\dot{s}'_{(1)} &= \left( \Pi(s'_{(1)}) - \langle s_1^\star, s'\rangle s'_{(1)} (s_1^\star)^\top \right) \left( \Pi(s'_{(1)}) - \langle s_1^\star, s'\rangle s_1^\star s_{(1)}'^\top \right) \left( V_{(1)}'^\top G_{(1)} - \|V'\|_F^2 s'_{(1)} \right)\\
&= \left( \Pi(s'_{(1)})^2 + \langle s_1^\star, s'\rangle^2 \|s_1^\star\|^2 s'_{(1)} s_{(1)}'^\top \right) \left( V_{(1)}'^\top G_{(1)} - \|V'\|_F^2 s'_{(1)} \right) \,.
\end{aligned}
$$

Similarly, writing the update for $V'_{(1)}$ and the update in the direction of $V_1^\star$:

$$
\begin{aligned}
\langle V_1^\star, \dot{V}'\rangle &= -\|s'_{(1)}\|^2 \langle V_1^\star, \dot{V}'\rangle \,,\\
\dot{V}'_{(1)} &= G_{(1)} s'_{(1)} - \|s'_{(1)}\|^2 V'_{(1)} \,.
\end{aligned}
$$

We are ready to state the main theorem:

**Theorem 4.** *Assume that the initialization satisfies the following for all $k \in [2, h]$:*

$$\langle V'(0), V_2^\star\rangle \ge \langle V'(0), V_k^\star\rangle \,, \quad \langle s'(0), s_2^\star\rangle \ge \langle s'(0), s_k^\star\rangle \,.$$

*Further, suppose that $V'(0), s'(0)$ are such that*

$$\langle V'_{(1)}(0), G_{(1)} s'_{(1)}(0)\rangle > \frac{1}{2}\|V'(0)\|_F^2 \|s'_{(1)}(0)\|^2 \,. \tag{10}$$

*Then, the dynamics of $V'$ and $s'$ converge to the following fixed point:*

$$V'(\infty) = m_2^\star V_2^\star \,, \quad s'(\infty) = s_2^\star \,.$$

*Proof.* We follow the same strategy as in Theorem [1]. Let $\mathcal{R}$ be the following set:

$$\mathcal{R} = \{(V', s') \mid \forall k \in [2, h], \langle V', V_2^\star \rangle \geq \langle V', V_k^\star \rangle \text{ and } \langle s', s_2^\star \rangle \geq \langle s', s_k^\star \rangle\}.$$

We prove that the flow is forward-invariant on $\mathcal{R}$.

Fix any $j \in [2, h]$. Let $w_j = \langle V', V_2^\star - V_j^\star \rangle$ and $z_j = \langle s'_{(1)}, s_2^\star - s_j^\star \rangle$. The flow of $w_j$ and $z_j$ are as follows:

$$\dot{w}_j = m_2^\star \langle s'_{(1)}, s_2^\star \rangle - m_j^\star \langle s'_{(1)}, s_j^\star \rangle - \|s'_{(1)}\|^2 w_j,$$

$$\dot{z}_j = (s_2^\star - s_j^\star)^\top \left( \Pi(s'_{(1)})^2 + \langle s_1^\star, s' \rangle^2 \|s_1^\star\|^2 s'_{(1)} s'^\top_{(1)} \right) \left( V'^\top \boldsymbol{G}_{(1)} - \|V'\|_F^2 s'_{(1)} \right).$$

Rewriting the derivative of $\dot{z}_j$:

$$\dot{z}_j = (s_2^\star - s_j^\star)^\top \Pi(s'_{(1)})^2 \left( V'^\top \boldsymbol{G}_{(1)} - \|V'\|_F^2 s'_{(1)} \right) + c z_j$$

$$= (s_2^\star - s_j^\star)^\top \operatorname{diag}(s'_{(1)})^2 \left( V'^\top \boldsymbol{G}_{(1)} - \|V'\|_F^2 s'_{(1)} \right)$$
$$- (s_2^\star - s_j^\star)^\top \operatorname{diag}(s'_{(1)}) s'_{(1)} s'^\top_{(1)} \left( V'^\top \boldsymbol{G}_{(1)} - \|V'\|_F^2 s'_{(1)} \right) + c z_j$$

$$= (s_2^\star - s_j^\star)^\top \operatorname{diag}(s'_{(1)})^2 \left( V'^\top \boldsymbol{G}_{(1)} - \|V'\|_F^2 s'_{(1)} \right)$$
$$- \left( \langle s'_{(1)}, s_2^\star - s_j^\star \rangle s_2^\star + \langle s'_{(1)}, s_j^\star \rangle (s_2^\star - s_j^\star) \right)^\top s'_{(1)} s'^\top_{(1)} \left( V'^\top \boldsymbol{G}_{(1)} - \|V'\|_F^2 s'_{(1)} \right) + c z_j$$

$$= m_2^\star \langle s_2^\star, s'_{(1)} \rangle^2 \|s_2^\star\|^2 \langle V', V_2^\star \rangle - m_j^\star \langle s_j^\star, s'_{(1)} \rangle^2 \|s_j^\star\|^2 \langle V', V_j^\star \rangle + c z_j,$$

where $c$ is some arbitrary time-dependent function that changes from line to line. On the boundary of $\mathcal{R}$, we have $w_j = 0$ or $z_j = 0$. If $w_j = 0$, then $\dot{w}_j \geq 0$ and if $z_j = 0$, then $\dot{z}_j \geq 0$. Therefore, a flow that has started in $\mathcal{R}$ will remain in $\mathcal{R}$ for all time.

Now, consider the following Lyapunov function:

$$\phi(V', s'_{(1)}) = \langle V', \boldsymbol{G}_{(1)} s'_{(1)} \rangle - \frac{1}{2} \|V'\|_F^2 \|s'_{(1)}\|^2.$$

The derivative of $\phi(V', s'_{(1)})$ is as follows:

$$\nabla_{V'} \phi(V', s'_{(1)}) = \boldsymbol{G}_{(1)} s'_{(1)} - \|s'_{(1)}\|^2 V',$$

$$\nabla_{s'_{(1)}} \phi(V', s'_{(1)}) = V'^\top \boldsymbol{G}_{(1)} - \|V'\|_F^2 s'_{(1)}.$$

Therefore, the time derivative of $\phi$:

$$\dot{\phi} = \|\dot{V}'\|^2 + \|\tilde{\Pi}(s') \nabla_{s'_{(1)}} \phi(V', s')\|^2 \geq 0,$$

where $\tilde{\Pi}(s')$ is a positive semi-definite matrix that satisfies:

$$\tilde{\Pi}(s')^2 = \left( \Pi(s'_{(1)})^2 + \langle s_1^\star, s' \rangle^2 \|s_1^\star\|^2 s'_{(1)} s'^\top_{(1)} \right), \quad \ker(\tilde{\Pi}(s')) \subseteq \ker(\Pi(s'_{(1)})).$$

By Equation (10),

$$\phi(0) = \phi(V'(0), s'_{(1)}(0)) > 0,$$

and $s'_{(1)} \neq 0$ as $\phi$ is increasing. $\phi$ is optimized when $V' = \dfrac{\boldsymbol{G}_{(1)} s'_{(1)}}{\|s'_{(1)}\|^2}$ which leads to a finite value upper bound on $\phi(V', s'_{(1)})$.

Therefore, $\lim_{t \to \infty} \phi(V'(t), s'_{(1)}(t))$ is finite and the flow converges to a stationary point of $\phi$. That is, the flow converges to a point $(V'_\infty, s'_\infty)$ that satisfies:

$$\boldsymbol{G}_{(1)} s'_\infty - \|s'_\infty\|^2 V'_\infty = 0, \quad V'^\top_\infty \boldsymbol{G}_{(1)} - \|V'_\infty\|_F^2 s'_\infty \in \ker(\Pi(s'_\infty)).$$

Note that we have the following equality:

$$(\boldsymbol{G}_{(1)}s_\infty)^\top \boldsymbol{G}_{(1)} = \sum_{j=2}^h m_j^\star \left\langle V_j^\star, \sum_{k=2}^h m_k^\star V_k^\star \langle s_k^\star, s_\infty' \rangle \right\rangle s_j^\star = \sum_{j=2}^h (m_j^\star)^2 \langle s_j^\star, s_\infty' \rangle s_j^\star \,.$$

Then, the stationary point $(V_\infty', s_\infty')$ satisfies

$$\sum_{j=2}^h (m_j^\star)^2 \langle s_j^\star, s_\infty' \rangle s_j^\star - h^2 \|s_\infty'\|^2 \|V_\infty'\|_F^2 \langle s_j^\star, s_\infty' \rangle s_j^\star \in \ker(\Pi(s_\infty')) \,.$$

We have proven that $\langle s_2^\star, s_\infty' \rangle > 0$ as $\langle s_2^\star, s_\infty' \rangle = \max_{k \in [2,h]} \langle s_k^\star, s_\infty' \rangle$. From Lemma 2, $s_2^\star \notin \ker(\Pi(s_\infty'))$. By projecting to the direction $s_2^\star$,

$$(m_2^\star)^2 \langle s_2^\star, s_\infty' \rangle - h^2 \|s_\infty'\|^2 \|V_\infty'\|^2 \langle s_2^\star, s_\infty' \rangle = 0 \,.$$

However, note that

$$h^2 \|s_\infty'\|^2 \|V_\infty'\|_F^2 = \frac{\|\boldsymbol{G}_{(1)} s_\infty'\|^2}{\|s_\infty'\|^2} \le \max_{\|s\|=1} \|\boldsymbol{G}_{(1)} s\|^2 = (m_2^\star)^2 \,,$$

with equality if and only if $s_\infty = s_2^\star$. Therefore, the flow converges to the stationary point

$$s = s_2^\star \,, \quad V = m_2^\star V_2^\star \,.$$

$\square$

Lastly, we justify the initialization assumption in Equation (10). Theorems 1 and 2 demonstrate that a wide range of symmetric initializations converge toward the configuration defined in Equation (13). Note that Equation (10) requires stronger alignment than Equation (13), specifically that the tensor factorization loss is strictly lower than the value attained at the first saddle point characterized by Theorem 4. In practice, this condition is satisfied by a small perturbation along the second positional feature:

**Remark 2.** *Equation* (10) *is satisfied by the following initialization*

$$V'(0) \approx \frac{m_1^\star}{h} V_1^\star + \epsilon V_2^\star \,, \quad s'(0) \approx (1 - \epsilon)s_1^\star + \epsilon s_2^\star \,,$$

*for small $\epsilon > 0$.*

### E.4. Extension to Higher-order Heads

As in Section E.3, we study the offshoot of an arbitrary head $n > 2$ after the system has learned the first $n - 1$ features. The features $2, 3, \ldots, n - 1$ are all learned by a single head, whereas the remaining $h - n$ heads are still aligned with the first feature. This leads to dynamics analogous to Equation (8):

$$V_1 = V_{n+1} = \ldots = V_h = V \,, \quad s_1 = s_{n+1} = \ldots = s_h = s_1^\star \,, \quad s_2 = s_2^\star, \ldots, s_{n-1} = s_{n-1}^\star \,.$$

We assume an analog of the initialization in Equation (13):

$$V(0) = \frac{1}{h - n + 1} \left(m_1^\star V_1^\star - \langle s_1^\star, s_n \rangle V_n(0)\right) \,,$$
$$V_i(0) = m_i^\star V_i^\star - \langle s_i^\star, s_n \rangle V_n(0) \,, \quad \forall i \in [2, n-1] \,,$$
$$V_n(0) \approx V(0) \,, \quad s_n(0) \approx s_1^\star \,.$$

This leads to similar dynamics after assuming that $V, V_2, \ldots, V_{n-1}$ have fast dynamics by an argument similar to Lemma 1. Here, we write $V' = V_n$ and $s' = s_n$ for brevity:

$$\dot{V}' = \boldsymbol{G}_{(n-1)} (s_n)_{(n-1)} - \|(s_n)_{(n-1)}\|^2 V_n \,,$$
$$\dot{s}' = \Pi(s_n)^2 \left(V_n^\top \boldsymbol{G}_{(n-1)} - \|V_n\|_F^2 (s_n)_{(n-1)}\right) \,.$$

Computing the update in the relevant directions of $s'$, we obtain:

$$\dot{s}'_{(n-1)} = \left( \Pi(s'_{(n-1)})^2 + \sum_{j=1}^{n-1} \langle s_j^\star, s' \rangle^2 \|s_j^\star\|^2 s'_{(n-1)} s'^\top_{(n-1)} \right) \left( V'^\top G_{(n-1)} - \|V'\|_F^2 s'_{(n-1)} \right) .$$

The same analysis in Section E.3.1 leads to the following theorem:

**Theorem 5.** *Assume that the initialization satisfies the following for all $k \in [n, h]$:*

$$\langle V_n(0), V_n^\star \rangle \geq \langle V_n(0), V_k^\star \rangle \quad \langle s_n(0), s_n^\star \rangle \geq \langle s_n(0), s_k^\star \rangle .$$

*Further, suppose that $V_n(0), s_n(0)$ are such that*

$$\langle V_n(0), G_{(n-1)}(s_n)_{(n-1)}(0) \rangle > \frac{1}{2} \|V_n(0)\|_F^2 \|(s_n)_{(n-1)}\|^2 .$$

*Then, the dynamics of $V_n$ and $s_n$ converge to the following fixed point:*

$$V_n(\infty) = V_n^\star , \quad s_n(\infty) = s_n^\star .$$

*Proof.* The proof proceeds mutatis mutandis to that of Theorem 4. $\square$

## F. Expanding the Initialization Condition

In this section, we explain Remark 1 in detail. As stated, for any initialization around $s_k(0) \approx \frac{1}{T}1_T$ and $V_k \approx 0$, we obtain the following from the first-order Taylor approximation as $P \approx 0$:

$$\dot{V}_k(0) \approx \frac{1}{T}G1_T , \quad \dot{s}_k(0) \approx 0 .$$

Therefore, the heads $V_k(0)$ exhibit a faster dynamics than the attention scores $s_k$. For small timescales $t$, the heads are approximately aligned with the same direction:

$$V_k(t) \approx \frac{t}{T}G1_T ,$$

which satisfies the initialization condition in Theorem 1 as $m_1^\star \geq m_k^\star$ for any $k \in [h]$. Moreover, the second-order Taylor approximation yields:

$$\ddot{V}_k(0) \approx -\sum_i \dot{V}_i s_i^\top s_k \approx \frac{1}{T^2}G1_T ,$$

$$\ddot{s}_k(0) \approx \Pi(s_k)\dot{V}_k^\top (G - P) \approx \frac{1}{T}\pi(s_k)G1_T .$$

By Lemma 3, we can show that $\dot{s}_k(0)$ is such that the component of $s_1^\star$ is the maximal entry. Therefore, we expect $s_k$ to align with the initialization condition given in Theorem 1 for small timescales $t$:

$$s_k(t) \approx \frac{1}{T^2} \left( I_T - \frac{1}{T}1_T 1_T^\top \right) G1_T .$$

A similar type of analysis also applies to the initializations of Theorems 4 and 5.

Note that the initialization regimes in our theorems are not near a particular point but large sets that satisfy some ordering. Coupled with the analysis above, the initialization basin for these theorems can be expanded. This contrasts with analyses that rely on vanishing initialization or limits toward critical submanifolds.

