# OpenReview forum: "Incremental Learning of Sparse Attention Patterns in Transformers"
_ICML.cc/2026/Conference — ICML 2026 regular_

### Official Review · Reviewer_H41G · 2026-02-28

**Soundness:** 2
**Presentation:** 3
**Significance:** 3
**Originality:** 3
**Overall Recommendation:** 4
**Confidence:** 3

**Summary:**

The paper studies how transformers learn high order Markov chains (MCs). In this setting, given an importance hierarchy between the MC constituents, those constituents are shown to be learned sequentially from the most to the least important. Furthermore it is observed that heads specialize, with each head taking on the task of one of the MC constituents; the authors name this the "cooperative" stage. It is also argued that early stopping / limited data biases the model toward simpler partial-context predictors. A different setting is studied theoretically and is shown to exhibit a qualitatively similar phenomenology.

**Compliance With Llm Reviewing Policy:**

Affirmed.

**Final Justification:**

As I detailed in my review and my comments, this paper overall suffers from framing issues and overclaiming. The theoretical part is also highly problematic, with only a loose connection to the experimental part; this is in stark contrast to how modeling is done in other sciences (e.g. physics) where theoretical assumptions are confirmed empirically, and there is a clear limit in which the assumptions are justified. That said, I think the experimental section is of interest to the community. I therefore recommend acceptance, but with some substantial reservations. Under an ideal peer-review process, I don't think this work should be accepted as is or under the changes the authors suggested.

**Key Questions For Authors:**

- How would the results change for number of heads different than number of MC constituents $k \neq h$?
- It seems the construction in 2.3 assumes $d_k = d_q > T$ which is commonly not the case in practice, how essential is this assumption experimentally?
- Assumption 1 seems to be incompatible with one-hot input, can you elaborate on that?
- Does the specific cooperative behavior mentioned in lines 364-369 (right) appear in the full model too?
- "In fact, we can derive a precise statement about how far V is from V ⋆ based on how much weight s′ puts on the directions that are orthogonal to s1" does this hold for the experimental model too? Approximately?
- Can the authors support and contextualize this claim in the introduction? References to some works could be appropriate here.
  "Transformers implement this operation across different positions via sparse attention patterns which pushes their parameters to diverge."
- "For simplicity, we choose αi = 1/|I(I−1(i))| where I−1 is the inverse of I." Do you mean that $\alpha$ is constant within each interval $I$ and the sum within the interval is normalized to $1$?
- How should the left and right panel of figure 3 be taken together?
- In lines 355-356 (right) $O(\eta)$ appears, what is $\eta$?

**Limitations:**

Granting the major declared simplifications of the theoretical model (last token regression instead of next token "classification"), the theoretical model remains quite far from the experimental one, and this gap is not discussed clearly enough.
- Several architectural elements often viewed as central to transformers are removed or altered in the theory setup, which makes it hard to interpret the proofs as explaining the observed behavior.
- The use of population gradient flow (rather than DMFT-like dataset averaged gradient flow or online SGD treatment) is an important limitation and should be surfaced earlier and emphasized more consistently; as written, it appears late and is easy to miss.
- More generally, a number of prominent claims in the abstract and “main contributions” read stronger than what is directly supported by the experiments/theory, which contributes to a mismatch between stated scope and demonstrated results.

**Strengths And Weaknesses:**

### Strengths
- The paper is clearly written and contextualized well.
- The experimental section shows interesting and clear results.
- The experiments are well designed and convincing of the main experimental claims (see a few exceptions below)
### Weaknesses
- **Modeling Choices:** Prima facie, some of the changes in the theoretically studied setting seems drastic:
	- Using position only attention makes it more similar to a (multi head/channel) MLP mixer. The fact that transformers don't have weights that explicitly act in sequence space directly gives rise to a different inductive biases and dynamics.
	- The theoretical results seem to be contingent on having fine-tuned initialization, though the experimental results are robust to the initialization scheme. To me, this indicated that the mechanism that drives the stage-wise learning in the experimental setup might be different than the one that drives it in the theoretical one.

	Both of these points can be overcome by showing that the theoretical results are able to predict quantities that are relevant to the experimental setup (even without proof). In this case I would be happy to recommend that paper be accepted.
- **Linking Theory and Experiment:** The theoretical results (Thm.1-4) are never plotted / compared with the experiments in this setting (Fig.10). I believe such comparison would strengthen this work. Is the *scaling* predicted by the results tight (ignoring constants)? Are the timescales in these results relevant? (Can the constants be determined?) Any sort of visualization or comparison of the theoretical results to their experiments in Fig.10 would strongly enhance this submission in my view. The same goes to comparing Fig.10 results to main text results.
  Considering my previous reservation from the simplification made in the theory I am concerned that staged learning is not a strong enough signature to claim the theory explains the observations (many simple theories may yield stage-wise learning).
- **Framing and Support of the Main Claims**: Some of the claims under "main contributions" are not substantiated in the text.
	- "In particular, we **isolate** the importance of sparse attention patterns as the driving force for incremental learning in transformers"
	  I don't see how the results isolate the importance of sparseness. Would the result in case of a (non-overlapping) dense attention pattern look differently? The paper never address this question as far as I can tell.
	- "We prove a convergence result that explains the behavior of the first competitive phase as a coupled dynamics **induced by the symmetry of the initialization.**"
	 The paper attributes the competitive phase to ‘symmetry of initialization.’ The theorems, however, require initialization within an $\epsilon$-neighborhood of the symmetric manifold to obtain coupling (for a time window depending on $\epsilon$). IID initialization gives exchangeability but not equality; the paper should either (i) justify that the relevant parameters/attentions concentrate so that $\epsilon$ is small in the experimental setup, or (ii) soften the claim. An empirical check, e.g., measuring head-to-head distances at initialization or early in training and comparing the observed coupling time against the theorem’s $\epsilon$-dependence would make this link convincing.
	- "This suggests that there is a regularization **induced by the training trajectories**, where transformers are pushed to be misspecified depending on the size of the training set." I believe the bolded part is not supported by this work. It seems the authors imply the specific training dynamics induces this bias, where it could also be a result of the architectural inductive bias. To support such claim one should show that it is not present in more naive algorithms like Guess&Check (or Bayesian sampling which is insensitive to the dynamics). The fact that adam and SGD shows similar behavior somewhat weakens this claim further.

	The statement of main contribution should be refined to reduce the gap between what is claimed and what is actually present in the work. I emphasize that I believe the problem is the statement of the contribution, not the content of the work itself.
- The abstract can also be read as over-claiming: "provides insights into the **emergence** of complex behaviors" emergence is often used in the literature to indicate an abrupt phenomena or a collective phenomena that does not exist for each individual part. The authors do not indicate this is the case here and do not discuss this point further. I note the manuscript itself uses 'emergence' to refer to the appearance of induction heads, a phenomena that is believed to be abrupt.

---

> ### Author Rebuttal · Authors · 2026-03-31
>
> We thank the reviewer for the detailed and constructive feedback.
>
> **W1, W2**
>
> We share **new experiments** on the following [anonymous repository](https://anonymous.4open.science/r/icml-2026-incremental-rebuttal-2843/).
>
> **Link between Figure 10 and theory.** We plot the theoretical trajectories alongside the full dynamics of the regression objective (Figure A). The theoretical trajectories capture the original trajectory with high fidelity; the accuracy of this connection is dependent on the initialization (see W3).
>
> **Link between transformers and Figure 10.** We plot value matrix alignment and attentions in the style of Figure 10 for the full transformer (Figure B). Qualitatively, the two plots are very similar: attention trajectories diverge incrementally, formation of feature matrices coincide with the bifurcations in attention, and cooperative behavior predicted by the theory is present in transformers.
>
> **Timescales.** Our theory derives convergence results rather than timescales, as the induced dynamics reduce to an extremely nonlinear ODE that is hard to analyze for our initialization of interest ($s \approx 1/T \mathbf{1}_T$).
>
> **W3**
>
> We will improve the text by incorporating the following points:
>
> * Edelman et al. (2024) first empirically showed saddle-to-saddle dynamics across multiple heads, where transformers learn copying circuits (1st layer) combined with an induction head (2nd layer). We hypothesized that the fundamental reason for the incremental learning is that copying requires sparse attention patterns, which leads to saddles due to diverging parameters (see Q6). By using the simplest possible statistical estimator (linear) on top of copying circuits, we _isolate_ the importance of these sparse patterns by removing the complex second layer of Edelman et al. (2024). Our work points at the generality of sparse attention formation as the key mechanism driving incremental learning beyond this specific setting.
>
> * The closeness of heads, of course, is affected by the initialization scale. Literature on incremental learning almost always requires the much more stricter assumption of _vanishing initalization_. For our problem, the attention patterns start at uniform without any initialization tricks (see Figures in the Appendix for $t=0$) and this can be verified with transformers in practice. However, our rigorous results pertain the small initialization regime for value matrices.
>
> * We did not want to claim that the training algorithm is the sole origin of the regularization. Our understanding is that gradient-based optimization in the landscape formed by the multi-head architecture exhibits interesting regularization behaviors. As the reviewer noted, understanding the source of this bias requires further investigation.
>
> * We relate our results to emergence as transitions between various phases occur rather abruptly, usually after longer plateaus.
>
> **Q1** We ran a preliminary experiment in Appendix B.7. In general, we expect the first $h$ features to be learned when $h \leq k$, and the extra $h - k$ heads to share the most important feature when $h > k$.
>
> **Q2** We used concatenated one-hot positional embeddings for simplicity. Representations with relative or sinusoidal positional encodings, which do not require $\mathcal{O}(T)$ dimensionality, can also be used and we have verified experimentally that the incremental learning behavior is the same.
>
> **Q3** Please see our response to W4 of Reviewer 8Fhk where we explain we are able to extend the covariance structure (including to any scale constrained by one-hot tokens).
>
> **Q4** Transformers do exhibit the cooperative behavior as predicted by the regression ODE (Figure B in W1, W2).
>
> **Q5** Lemma 1 qualitatively captures the dynamics in our simulations (Figure A in W1, W2).
>
> **Q6** This follows from the positivity of the softmax: sparse attention patterns require the logits to grow without bound. By sparse attention patterns, we think of patterns where some entries are exactly zero, which necessities a scale separation between zeros and non-zeros. See also Nichani et al. (2024) and Edelman et al. (2024).
>
> **Q7** Yes, $\alpha_i$ is constant within each interval $I(k)$ and normalized so that the sum within each interval equals 1. We explore non-uniform $\alpha$ values in Appendix A.3, where incremental learning is still observed.
>
> **Q8** The left compares the transformer to the hand-crafted partial predictors in Equation 3, while the right compares to fully trained transformers with restricted context lengths. Both show the same distinct stages, with small differences arising because the transformer's parameters simultaneously adjust to accommodate all patterns during training.
>
> **Q9** $\Pi(s') = \mathcal{O}(|s' - s_1^\star|) \approx 0$, indicating that the projection operator is small when $s'$ is close to $s_1^\star$.
>
> **Limitations.** We will add a dedicated paragraph to explicitly acknowledge listed limitations.

---

> > ### Author Rebuttal · Reviewer_H41G · 2026-04-02
> >
> > I thank the authors for their clear rebuttal and new experiments.
> > I find figure A compelling and decide the soundness score to 2. Below I detail what I think is still missing in the current submission.
> >
> > **Linking Theory and Experiment.** As I noted in my original review, I do not believe that staged learning is a strong enough signature to claim the theory explains the observations (many simple theories may yield stage-wise learning). For example the time scales in figures A and B are very different and its hard to identify any feature that carry over from figure A to figure B.
> >
> > **Framing and Support of the Main Claim.** The response fails to address my concerns. I see ensuring the statement of main contributions reflects the content of the paper as a requirement for acceptance.
> > 1. My main concern remains: Would the result in case of a (non-overlapping) dense attention pattern look differently? The paper never address this question as far as I can tell. The rebuttal largely addressed an aspects that was never in question (whether it is the first or second layer), what I take issue with is that the authors credit sparseness and claim they "isolate its effect. Isolating its effect is not possible without addressing the counter hypothesis that it is irrelevant.
> > 1. My problem with the second and third main contributions are largely the same: The paper establishes A->B but the authors claim A<->B. The work shows that given "symmetry of the initialization" they can explain "the behavior of the first competitive phase", the work does not show that "symmetry of the initialization" is a necessary condition for the competitive behavior. Therefore the work does not establish "symmetry of the initialization" as the cause of the observed behavior.
> >
> > I note that I believe the required changes are rather minor, but nevertheless are important. If the authors can rephrase the main contributions such that they fully align with the content of the paper I will increase my overall score.
> >
> > Questions 1,2,3,4,5,8,9 weren't fully addressed, but the authors should prioritize the two points I mentioned above.

---

> > > ### Author Response · Authors · 2026-04-06
> > >
> > > We thank the reviewer for the continued engagement and for clarifying exactly what is needed for acceptance. We now address the two points more fully.
> > >
> > > **Linking Theory and Experiment.** We want to clarify what we intended Figures A and B to demonstrate. The two figures are from fundamentally different settings: Figure A shows the regression model from Section 3 (a simplified architecture, population gradient flow), while Figure B shows the full transformer trained on the classification task. We never intended them to be quantitatively matched: the dimensions, the scales of the norms, the y-axis values, and the timescales are all different by construction, and no attempt was made to align any of these (both figures use the same hyperparameters as the main paper).
> > >
> > > What we intended to highlight is that the structural similarities between the two figures go well beyond staged learning. Concretely, Figure B reproduces the following features predicted by the theory: (i) the competitive phase, where all heads initially converge toward the most important position (Position 3); (ii) the sequential offshoot structure, where heads leave one by one: Head 2 first acquires Position 2 before Head 3 acquires Position 1; (iii) the collaborative mechanism at each stage: during Stage 2, Heads 1 and 3 decrease their alignment with Feature 2 to offset the residual of Head 2 in Position 3, and during Stage 3, Head 3 decreases its alignment with Feature 1 to offset the residual of Head 1. These are non-trivial structural signatures specific to our theoretical mechanism, not properties shared by any stage-wise learning model. We note that we only checked for the compensatory decrease in value matrix alignment following the reviewer's question, and this check revealed that the theory is indeed qualitatively predictive in ways we had not previously verified.
> > >
> > > Given that the dynamics of full transformers are highly nonlinear and involve many components absent from our theory, we are happy to explicitly acknowledge the limitations of the match between theory and experiments. We believe, however, that the structural agreement described above constitutes meaningful qualitative evidence that the theory points toward the right mechanism and we will make the scope of our claim and framing clear in the revision.
> > >
> > > **Framing and Support of the Main Claim.**
> > >
> > > We fully agree and will revise the phrasing throughout to align with the content of the paper, as detailed below.
> > >
> > > **Sparse attention patterns.** When we wrote "isolate the importance of sparse attention patterns," we meant isolate relative to Edelman et al. (2024)'s setup, where a second layer is needed to form induction heads. The reviewer is correct to flag this: we believe sparseness is a key feature driving incremental learning, but we have not proved it is the only mechanism through which incremental learning can occur. Regarding the non-overlapping dense attention counterfactual: we hinted at this in Q6. Any attention pattern where the softmax concentrates on disjoint subsets of positions is sparse in our sense (the attended positions may form a large block rather than a few positions, but the unattended positions still require logits to diverge), which produces the same saddle structure.
> > >
> > > We will revise the language to make this notion of sparseness explicit throughout the paper and replace "isolate the importance of sparse attention patterns as the driving force" with "identify sparse attention patterns as a driver of incremental learning."
> > >
> > > **Symmetry of initialization.** We never intended to claim that symmetry is a necessary condition for competitive behavior. Our point was that near-symmetric initialization leads to a tractable ODE whose solution qualitatively matches the experiments: all heads compete to acquire the first sparse attention pattern. The theorem establishes a sufficient condition, not a characterization. As we argued, one satisfies the conditions of this theorem when value matrices are initialized small (as attention patterns start uniform by default). Figure 4 (left) shows that the incremental learning is indeed more pronounced with smaller initialization scales.
> > >
> > > We will replace "explains the behavior of the first competitive phase as a coupled dynamics induced by the symmetry of the initialization" with "provides a sufficient condition for the competitive phase: near-symmetric initialization induces a coupled dynamics that converges to the same fixed point as qualitatively observed experimentally."
> > >
> > > **Regularization induced by training trajectories.** As noted in our original rebuttal, we did not intend to claim that the optimizer is the sole source of this regularization. We will replace "a regularization induced by the training trajectories" with "a regularization effect associated with the trajectory of the high-data regime full run."

---

### Official Review · Reviewer_Hchg · 2026-03-10

**Soundness:** 3
**Presentation:** 2
**Significance:** 2
**Originality:** 2
**Overall Recommendation:** 4
**Confidence:** 4

**Summary:**

This paper studies how Transformers learn on a synthetic high-order Markov chain task, where prediction requires using several past tokens with different importance.
The authors find that learning happens in stages. At first, attention heads compete and focus on the most important pattern. Later, they specialize and cooperate, with different heads attending to different positions through sparse attention patterns.
They explain this behavior with a simple theoretical model of the training dynamics, showing that the transformer gradually learns more complex representations over time.

**Compliance With Llm Reviewing Policy:**

Affirmed.

**Final Justification:**

I have increased my score from Weak Reject to Weak Accept following the author's rebuttal and the subsequent discussion. While I initially had concerns regarding the predictability of the results and the practical significance of the toy task, the authors have effectively addressed these points and clarified the unique theoretical contributions of their work.

**Key Questions For Authors:**

- This toy task captures positional importance in sequential data. Which properties of natural language are not captured by this setup? For example, a hierarchical structure.

- Transformers are widely used beyond language (vision, time-series, etc.).
Do you expect the same competitive-to-cooperative head dynamics to appear in those domains as well?

**Limitations:**

yes

**Strengths And Weaknesses:**

**Strengths**
1. Interesting learning dynamics observation

The observation that attention heads initially compete for the most important feature and later cooperate by specializing into different patterns is interesting. The empirical visualization of training trajectories clearly illustrates these stages.

2. Strong theoretical analysis

The paper provides a relatively rigorous theoretical explanation using simplified differential equations and tensor factorization perspectives. This helps connect empirical training dynamics with interpretable mathematical structure.

3. Clean toy problem

The high-order Markov chain setup isolates positional dependencies and different importance levels across past tokens. This provides a controlled setting to study the emergence of sparse attention circuits.


---

**Weaknesses**

1. Some results feel somewhat expected

The main empirical finding, that attention heads learn patterns in order of statistical importance, can feel somewhat intuitive given the task design. Since the next-token prediction depends on groups of positions with different importance weights, it seems natural that the model learns the most important signal first and gradually incorporates weaker signals.

Therefore, while the analysis is interesting, the qualitative outcome itself may appear somewhat predictable.

2. Writing clarity issues

The introduction could be improved in several ways:

It is unclear early in the paper what gap in prior work this paper fills.

- The task setup is difficult to understand when first reading the introduction.

- Referring to Figure 1 earlier and explaining the task more concretely would help the reader.

- In Figure 1, it is not immediately clear what $A_k^*$​ represents or how it relates to the prediction process.

Improving these aspects would significantly increase accessibility.

3. Limited discussion of practical implications

While the toy task captures some characteristics of natural language (e.g., positional importance and compositional dependencies), the paper does not clearly discuss how these findings translate to real-world transformer behavior.

For example, work such as [1] studies similar phenomena (competition and specialization of heads) and connects them to practical issues such as hallucination or instability. Even if such work appeared after this paper, discussing potential implications in real-world LLM settings would strengthen the contribution.

[1] Chakrabarti et. al., "Multi-Head Attention is a Multi-Player Game".

---

> ### Author Rebuttal · Authors · 2026-03-31
>
> We thank the reviewer for their constructive feedback and for recognizing our work as having a "clean toy problem" with "strong theoretical analysis" and "interesting learning dynamics". Below, we address your specific concerns and outline the changes we will make to the manuscript to resolve them.
>
> **W1: Some results feel somewhat expected**
>
> We agree the macroscopic outcome is intuitive, but our primary contribution is not _**what**_ is learned but _**how**_, and we believe the microscopic mechanism is genuinely non-trivial.
>
> Consider the competitive phase: a naive prediction would be heads to immediately divide labor. Instead, all heads simultaneously collapse onto the same pattern, as established in Theorem 1: under near-symmetric initialization, every head converges to the same fixed point with value $m_1^\star / h$.
>
> The cooperative phase is similarly non-obvious: The offshooting head's value matrix $V'$ grows orthogonal to $V_1^\star$ while the ensemble compensates by learning $-V_\perp$, a mechanism that emerges from the saddle-to-saddle dynamics. We will revise the introduction to foreground these contributions.
>
> **W2: Writing clarity**
>
> We agree the gap should be stated more precisely. \citet{edelman2024evolution} empirically observed that transformer heads learn in stages on n-gram tasks, with heads progressively acquiring more complex patterns. However, no prior work provides a theoretical explanation for why this happens at the level of individual head dynamics. We fill this gap by providing the first theoretical characterization of the full saddle-to-saddle multi-head dynamics. Technically, \citet{zucchet2025emergence} studies the most closely related mathematical setting ($h=1$, local escape from initialization), but the phenomena we analyze, head competition and cooperation, do not exist in the single-head case and our analysis is considerably more involved. We will update the introduction to make this distinction explicit.
>
> We will also make the following targeted changes to improve accessibility: (i) add a direct reference to the Figure 1 in the paragraph introducing our task; (ii) extend the Figure 1 caption to clarify that $A_k^\star \in \mathbb{R}^{d \times d}$ maps aggregated token representations to prediction logits, with larger $|A_k^\star|$ indicating greater influence on the output.
>
> **W3: Practical implications**
>
> We thank the reviewer for pointing us to [1], which we were not aware of. Our work characterizes a specific temporal unfolding of this multi-player game, from competitive convergence to cooperative specialization, and we will add a discussion connecting our results to their framing.
>
> More broadly, our results have two practical implications we will make more explicit in the paper. First, the abrupt transitions between stages correspond directly to sudden capability gains observed in transformer training \citep{chen2024sudden}: our theory provides a mechanistic explanation for why these emergent jumps occur in terms of saddle-to-saddle dynamics. Notably, the competitive-to-cooperative transition governs when each capability is acquired, suggesting that this phase structure may have consequences for the timing and ordering of emergent capabilities in practice. Second, our generalization experiments (Section 3.4) show that transformers trained on smaller datasets automatically stop at simpler hypothesis classes, pointing to a practically useful implicit regularization effect of the training trajectory.
>
> **Q1: Properties of natural language not captured?**
>
> Our task captures a horizontal structure, acquiring patterns at the same level of abstraction in order of importance. Natural language also has a vertical, hierarchical structure (words → phrases → sentences), where higher-level abstractions build on lower-level ones. Extending our framework to capture this type of incremental learning, where heads at different layers cooperate across levels of abstraction, is a natural and interesting direction for future work.
>
> **Q2: Beyond language**
>
> Even though we focused on language, we expect the same competitive-to-cooperative dynamics to arise in any domain where multiple attention heads are needed and there is an importance ordering over the patterns to learn. In vision transformers, nearby patches are more strongly correlated, providing a natural importance hierarchy analogous to our setting. Similarly, time-series forecasting with decaying autocorrelation across lags maps directly onto our task structure.

---

> > ### Author Rebuttal · Reviewer_Hchg · 2026-04-02
> >
> > Thank you for the clarification, but I believe this concern is not fully resolved.
> >
> > **> W3 (Practical implications)**
> >
> > First, the rebuttal refers to Section 3.4, but I could not find this section in the paper. Could you clarify which results you are referring to?
> >
> > Second, could you elaborate on how these findings translate to practical transformers, i.e., models trained on real-world natural language data? In particular, how should we interpret the connection to emergence or generalization beyond the synthetic Markov setting studied here?

---

> > > ### Author Response · Authors · 2026-04-06
> > >
> > > Thank you for the follow-up.
> > >
> > > **On Section 3.4**: We apologize for the confusion. This was a clerical error on our part: when entering the rebuttal on the submission website, we forgot to remove LaTeX citation commands and also misstated the section number. The generalization experiments we were referring to are in Section 2.5 ("Dataset Size and Generalization").
> > >
> > > Section 2.5 shows that as the dataset size decreases past critical thresholds, the number of learned stages drops: with less data, the model stops earlier in the incremental learning trajectory and converges to a misspecified model that copies only from a subset of the relevant positions (e.g., only $I(1)$ instead of $I(1) \cup I(2) \cup I(3)$). With early stopping, this misspecification is selected automatically, suggesting an implicit regularization where the model's effective context length is influenced by the amount of training data.
> > >
> > > **On the connection to practical transformers**: To summarize our position: (i) our contribution is a mathematical characterization of incremental learning in a controlled setting and we do not provide empirical evidence that our findings carry over directly to practical real-world LLMs; (ii) incremental learning has attracted interest both from the empirical side, where stage-wise capability acquisition has been documented in real-world LLMs (e.g., Chen et al., 2024), and from the theoretical side, where works such as Abbe et al. (2023) study the mechanisms underlying these stages in simplified settings. More generally, incremental learning and training dynamics of neural networks form a field with a large body of literature where empirical observations and theory have long informed each other.
> > >
> > > Our paper contributes to theoretical side: we establish a clean mathematical setting where the phenomena are unambiguous and the analysis is rigorous, with the goal of building intuition for the more complex real-world case. Within this scope, we can be precise about what our results do and do not say. What we can claim directly: the saddle-to-saddle dynamics we characterize provide a mechanistic account of abrupt capability jumps in a controlled setting, giving a concrete mathematical example of how "emergent" capabilities arise from smooth gradient dynamics passing through a sequence of saddle points rather than any fundamentally discontinuous process. What requires extrapolation: whether the same competitive-to-cooperative head dynamics appear in large-scale LLMs trained on natural language is not something our theory directly establishes. Real transformers have many layers, non-linearities, and learn over far more complex data distributions. The connection is qualitative: our results suggest that wherever attention heads must learn patterns with an implicit importance ordering, one might expect similar stage-wise dynamics. The generalization finding (smaller data → fewer stages → shorter effective context) is consistent with empirical observations that LLMs trained on less data fail to leverage long-range context, though we do not claim a direct causal link. Establishing such links via experimental evidence is important future work, but is outside the scope of the current paper, which studies incremental learning in a simplified and controlled setting.
> > >
> > > We will incorporate these clarifications into the manuscript. We hope this clarifies our position and the scope of our contribution.

---

### Official Review · Reviewer_8Fhk · 2026-03-12

**Soundness:** 3
**Presentation:** 2
**Significance:** 3
**Originality:** 4
**Overall Recommendation:** 4
**Confidence:** 4

**Summary:**

The paper considers a simple task based on Markov chains with importance structure to check the ability of transformers to implement multiple sparse attention patterns. It shows the essence of incremental learning in transformers. The paper provides both theoretical and experimental results for analyzing the learning dynamics of the multiple heads. The authors split the learning dynamics into two stages. The first stage is competitive, where all the heads try to learn the most important pattern. The second stage is cooperative, where the heads learn to predict the other patterns.

**Compliance With Llm Reviewing Policy:**

Affirmed.

**Final Justification:**

The rebuttal has addressed my main concerns. I will raise my score to 4.

**Key Questions For Authors:**

See weaknesses.

**Limitations:**

The authors did not discuss the limitations.

**Strengths And Weaknesses:**

Strengths:
1. The paper is well-motivated, and the task considered is designed clearly.
2. The paper investigates the learning dynamics from both the theoretical and empirical perspectives, making the analysis comprehensive.
3. It presents solid and clear mathematical analysis and proofs.


Weaknesses:
1. In Figure 4, it has two stages for $m=1$, while it has three stages for $m=1.3,1.5$. It is better to explain this difference.
2. In Theorem 1, it needs an infinite time to converge, and this result can not become the initial phase of the second stage. The initialization in Section D.1 and D.2 is too rigorous, with the exact form of the initialization of $V$ and $s$.
3. The paper is limited to a simplified, single-layer transformer model. It remains unclear how the results would extend to models with more layers or more heads.
4. Assumption 1 and Assumption 2 seem strong, especially for the assumptions about "normalized" and "orthogonal". Maybe the authors can emphasize the necessity of these assumptions in the paper.
5. In Section 2.1, it is said that the task considered is a classification task; however, as shown in eq(5), the authors utilize the mean square loss. This selection of the loss function may be a little strange.
6. In the paper, the empirical results are mixed with the problem setup, making the analysis of the experiments not so clear. It would benefit from clearly separating the task definition from the empirical results. In addition, some experimental details should be put in the paper.

---

> ### Author Rebuttal · Authors · 2026-03-31
>
> We thank the reviewer for the careful and detailed reading, and for recognizing the originality and soundness of our contributions. The weaknesses raised are largely about presentation and specific technical points, which we address below.
>
> **W1: Figure 4**
>
> We agree the caption should be more complete. The number of distinct phases depends on the importance gap $m$ and the alignment of the random initialization. When $m > 1$, the importance gap is large enough that the initialization is attracted to each saddle sequentially, yielding three distinct stages. When $m = 1$ there is no importance ordering, so the initialization is not preferentially attracted to any particular saddle — the model first learns one pattern due to random symmetry breaking, and then the remaining two are acquired simultaneously, collapsing to two stages. We will update the caption accordingly.
>
> **W2: Theorem 1 / initialization**
>
> As standard in the saddle-to-saddle literature, convergence results characterize the limit of each phase under idealized dynamics; the actual system passes through a neighborhood of each saddle before the next phase begins. Theorem 1 characterizes the limit under the coupled dynamics; Theorem 2 establishes that heads initialized near the symmetric point remain close to these coupled dynamics for a finite, controlled time, operationalizing the transition. Together they account for the competitive dynamics in our experiments. The cooperative phase analysis then takes over in the neighborhood of the Theorem 1 fixed point, as signaled by the $\approx$ in Eq. (9).
>
> Each phase leaves all heads approximately coupled at its end; given this near-symmetry, we analyze the idealized coupled initialization in Sections D.1 and D.2. Theorem 3 and its analogs provide the same style of stability bound, controlling deviations for each stage. Without the near-symmetric initialization assumption, the system is a highly complex nonlinear ODE where sharp convergence results are very difficult to obtain. Analyzing the dynamics in the coupled regime is a classical approach in this literature to make the overall nonlinear ODE tractable. We will add a clarifying remark in the main text making this chain of reasoning explicit.
>
> **W3: Limited to single-layer**
>
> We acknowledge this limitation. Note that a single attention layer is sufficient for the Markov chain task we study as no hierarchical composition is needed. Extending to multi-layer transformers would also require considering more complex input distributions where hierarchical processing is necessary. That said,  characterizing how this phase structure generalizes to deeper networks is an important future direction, and we will add a limitations paragraph.
>
> **W4: Assumptions 1 and 2**
>
> Assumption 2 can be safely removed, but the ordering of features must be handled more carefully: the condition $m_1^\star > m_2^\star > \cdots$ alone does not correctly capture relative importance when the feature matrices are not orthogonal, since cross-terms can arise: e.g., $V \otimes (s_2^\star + s_3^\star)$ may be more important than $V_1^\star \otimes s_1^\star$. The structure of the analysis remains intact with a refined ordering criterion based on the target tensor geometry.
>
> Relaxing Assumption 1 to a separable covariance $\mathbb{E}[x_i x_j^\top] = \gamma_{ij} \Sigma$ is also compatible with our analysis. Under this assumption, the gradient flow acquires two coupled geometries: a $\Sigma$-weighted inner product on feature matrices and a $\Gamma$-weighted inner product on attention vectors. The tensor factorization structure is preserved, and the dominant directions in each geometry must be defined accordingly. Beyond the separable form, a fully general per-position covariance $\mathbb{E}[x_i x_j^\top] = \Sigma_{ij}$ breaks the tensor factorization entirely, as the loss no longer decouples into separate feature and attention components.
>
> Finally, these relaxations change the problem to the form $\frac{1}{2}|GX - PX|_F^2$ rather than $\frac{1}{2}|G - P|_F^2$, i.e., a matrix sensing problem that reduces to pure matrix factorization only under isotropic covariance (Assumption 1). The sensing formulation is strictly harder, and whitening assumptions are standard in this literature to avoid this issue, see [Bach (2025)](https://francisbach.com/closed-form-dynamics/. We will clarify these in the paper.
>
> **W5: Loss function**
>
> As Reviewer rT1q also notes, simplified losses for theory are standard. Section 3 uses MSE loss to keep gradient flow tractable; cross-entropy on softmax outputs leads to intractable dynamics.
>
> **W6: Empirical results mixed with setup**
>
> We agree this affects readability. We will move Section 2.3 (the representation result) to directly follow Section 2.1, so the task definition and the construction are presented together as a self-contained formulation before any empirical results appear. We will also add explicit pointers to the experimental details in the Appendix.

---

> > ### Author Rebuttal · Reviewer_8Fhk · 2026-04-04
> >
> > Thank you for your response. I will raise my score to 4.

---

### Official Review · Reviewer_rT1q · 2026-03-12

**Soundness:** 2
**Presentation:** 3
**Significance:** 2
**Originality:** 3
**Overall Recommendation:** 4
**Confidence:** 3

**Summary:**

In the manuscript “12503 Incremental Learning of Sparse Attention Patterns in Transformers” the authors investigate the learning dynamics of Transformers trained on high-order Markov chains. They specify a toy-setting where contiguous segments of tokens each have unique importance and train both standard and simplified Transformer models on next-token prediction and regression variants of the task.

Empirically, they show that the learning exhibits stage-wise dynamics characterised by competition and cooperation. The attention heads initially _all_ focus on the most important parts of the sequence, whereafter heads gradually break away to re-prioritise on the next-most important parts.

The simplified Transformer model on the MSE task and some key assumptions allow them to treat the gradient analysis as a tensor-factorization problem, which has known convergence behaviours. They use this to characterise the competitive behaviour as coupled dynamics and show that this is caused by symmetry in the initialization, as all attention patterns and value matrices co-evolve. Post-convergence to the competitive fixed-point, instability around the saddle point amplifies differences in initialization and causes heads to break off to “re-focus” on different parts of the pattern. They demonstrate that these dynamics are entirely predictable from the symmetry of the initialization.

Finally, they demonstrate that reduced dataset diversity leads to fewer learning stages. In the most extreme examples, training a model on only 600 data points (for the same amount of training steps), corresponds to no learning stages at all. They make the claim that dataset size acts like implicit regularization towards simpler models because there are too few data points to learn the weaker correlations. As a result, the model itself chooses the right level of complexity for the available data.

**Compliance With Llm Reviewing Policy:**

Affirmed.

**Final Justification:**

This paper is a nice contribution.  The authors provide sufficient empirical evidence of their claims.  I recommend publication.

**Key Questions For Authors:**

1.  Do you have any sense of the robustness of these findings? How much of the behaviour survives if you relax the constraints on the data-generator? I’m mainly curious about whether there’s some setting that makes slightly fewer simplifying assumptions about the data and still convincingly shows the dynamics.

2.  What would we expect to happen in the scenario where multiple positions are equally important rather than having a strictly increasing/decreasing importance hierarchy? I think it’s relevant to consider the scenario where there are multiple paths that all greedily lower loss.

3.  Can you elucidate the storyline around low-data regimes acting as implicit regularization? Currently that storyline feels incomplete and seems like more of a remark rather than a meaningful contribution to the paper. It’s possible I’ve missed something, but it seems like a rather unsurprising result that models can overfit to small datasets.

**Limitations:**

yes

**Strengths And Weaknesses:**

Soundness:

The paper investigates their hypotheses both theoretically and empirically. The empirical results have sensible experimental setup and their results are interpretable and replicable. The setup is sensible for understanding how attention-patterns form.

The contiguous-block weighted data structure is perhaps less justified. Dependencies between data are often sparse but not contiguous. As the theoretical analysis does not depend on contiguousness of subsequences, it might be beneficial to investigate other sequence structures. The claim that the sequences are inspired by natural language is not justified, and the contiguous block structure conflates proximity and importance. The comparison with Transformers trained on shorter subsequence tasks holds only in this setting of contiguity and not in general. This is partly addressed in the appendix with inverse-order importance and overlapping groups, and it is firmly established within the literature that position-based attention patterns do emerge in stage-wise fashion and are largely found in real models. The agreement between the behaviour predicted by the theoretical work and what is observed in practice increases confidence in the applicability of this to real models.

The theoretical work makes reasonable assumptions. While it would be “nice” to keep the setting consistent with regular next-token prediction, the type of simplifications made to make the analytical work tractable is consistent with similar procedures carried out in the literature. However, this still leaves quite a bit of way between the main claim of the paper and the theory.

The paper’s remark that dataset size acts as being implicit regularization is fine but also carefully hedged. It is essentially a comment on overfitting where the early stopping point coincides with learning a simpler, misspecified hypothesis. I’m unsure whether we should expect this claim to generalise. The two things seem independent – could we expect the model to learn e.g. part of a hypothesis?

I’d like to commend the authors on the description of the experimental details. I was able to replicate all of their empirical results with no difficulty.

Presentation:

Overall the presentation is fairly good.

I find the narrative around the KL divergences between the full model and a sub-model trained on limited context length to be slightly confusing. The paper discusses the model learning the patterns incrementally but the plots show the model transition through a sequence of neighbourhoods where KL divergence eventually goes back up as the model transitions from one sub-model to the other. I understand what the authors are demonstrating but I find this particular presentation slightly unintuitive in how it communicates the story.

The relevant theory and results derived therefrom are laid out well, and the notation is consistent and easy to understand. However, the story becomes less clear once we get into the theory from section 3.0, and it gets harder to tell which results are from the simplified, theoretical setting and which are observed empirically on the full model.

Significance:

The topic is highly significant and training dynamics as a research direction is garnering significant attention within both the wider deep learning community and the field of interpretability.

While they do make several relaxing assumptions to make the problem tractable to study, they observe phenomena on a trained model with the full architecture consistent with their theoretical predictions. The main limiter of significance is lack of demonstration in models trained on real data, but given that this is theoretical work, this is to be expected. While the experiments in the main body of the paper have a narrow scope, the authors perform substantial extensions to both the theory and empirical work in the appendix which increases coverage. Importantly, the type of circuits they discuss do arise naturally in real language models.

Originality:

The training task is not novel and the stagewise learning dynamics and emergence of position-specific attention patterns have been studied before. However, the authors use a simpler training task such that they can more specifically isolate the formation of the attention patterns, and the subsequence specific weighting does allow them to derive insights that are absent from previous work. Notably, the analysis they perform is applicable to the multi-head Transformer setting and specifically focuses on the dynamics that occur between heads. The observed and derived results feel relevant and non-trivial, and as a predominantly theory-based paper that seeks to effectively isolate one mechanism of learning, it is a strong contribution. The authors show empirically that the results they derive theoretically are observable across different optimizers, parameters, and dataset sizes. It is clearly an extention of previous work but also meaningfully a contribution.

---

> ### Author Rebuttal · Authors · 2026-03-31
>
> We thank the reviewer for the careful, thorough reading and the positive overall assessment. We address each concern below.
>
> **Q1: Robustness under relaxed data-generator constraints**
>
> We expect the core behavior (competition followed by cooperation) to survive in settings with non-contiguous groups, any temporal ordering, and non-uniform attention weights within blocks. The key driver is the importance hierarchy between features, not the specific block structure. We point to several experiments in the appendix that support this:
>
> * Reversed importance ordering (Appendix A.2): the most important interval is the furthest group, breaking the association between proximity and importance. Incremental learning is still observed.
> * Non-uniform $\alpha$ values (Appendix A.3): within-block attention is non-uniform ($\alpha = [0.7, 0.3]$). The model still learns incrementally, with heads focusing proportionally more on the higher-$\alpha$ position.
> * Overlapping intervals with 3 and 4 heads (Appendix A.4): intervals overlap and there are more heads than features. Incremental learning persists, but the attention patterns for 3 and 4 heads show different orderings and different sparsity structures, illustrating that in more general settings the initialization and feature matrices jointly determine which sparse solution is selected.
>
> We note that in such settings, particularly with overlapping or non-disjoint groups, there are many possible sparse solutions, and the initialization may play a decisive role in which is selected. We further comment on the role of initialization in Q2. We will make this point explicit in the revision.
>
> **Q2: Multiple positions of equal importance**
>
> Stage-wise dynamics depend on both the initialization and the feature importances. More generally, the number of stages depends on the particular setting: when multiple features are equally important or when the importance gaps are small, several features can be acquired simultaneously, resulting in fewer visible stages. Figure 3 (right) shows one such example with $m=1$, where the model first learns a single pattern and then acquires the remaining two simultaneously.
>
> In fact, one can derive analogous results for such extreme cases from our theory: (i) all features learned simultaneously, or (ii) features learned strictly one by one, by assuming different initializations. We opted to present the full incremental dynamics as it is the more natural regime when there are clear differences in feature importance.
>
>
> **Q3: Low-data regime as implicit regularization**
>
> We acknowledge that the text was insufficiently precise on what kind of overfitting is occurring here, and we will include further discussion and attention visualizations in the revision to make this clearer. The key point is not just that the model overfits to small datasets, but that the **structure of the overfitting** is governed by the feature hierarchy imposed by the training dynamics.
>
> For example, with $n=600$ samples the model ends up between the first and second complexity levels, i.e., it has learned the two most important patterns but not the third. The cutoff point in the hierarchy is determined by the dataset size. The overfitting happens entirely within this subspace: the model fits the top-$k$ features and ignores the remainder, rather than partially fitting all features to varying degrees. This structured overfitting is a consequence of the incremental training trajectory and would not arise in a convex model such as linear regression, where overfitting distributes across all features without such a hierarchy. The training dynamics thus determine not just whether the model overfits, but which features it overfits to.
>
> Crucially, overfitting to the most statistically important features is the best possible outcome given limited data: early stopping regularizes the model at exactly the right level of complexity for what is learnable from the available samples, without any explicit model selection. This yields sample complexity benefits over a model that tries to learn everything.
>
> **Contiguous block structure:**
>
> We agree that contiguous blocks are one particular choice and do not claim it is the only or most general structure. The theory does not require contiguous blocks and the sets $I(k)$ can be arbitrary disjoint subsets of positions. The experiments in the Appendix with reversed importance ordering and varying interval lengths confirm that the incremental learning phenomenon is not specific to this choice. We will soften the natural language motivation in the revision to make clear that contiguous blocks are presented as one natural example rather than a direct model of language structure.

---

> > ### Author Rebuttal · Reviewer_rT1q · 2026-04-03
> >
> > I thank the authors for their clarifications. I find their arguments convincing and while more experiments to some extent can always be beneficial, I think they have sufficiently provided empirical evidence of their claims. I recommend publication.

---

### Decision · Program_Chairs · 2026-04-30

**Decision:**

Accept (regular)

**Comment:**

The paper studies single-layer multi-head transformers trained on a high-order Markov chain task where the next token depends on multiple past positions with different importance. Mirroring prior work by Edelman et al.'24, the central empirical claim is that learning proceeds in discrete stages, each marked by a head acquiring a sparse attention pattern over one group of informative positions. The paper studies these dynamics using simplified ODEs and prove stage-wise convergence. They show a compete vs cooperative dynamics, where heads first learn the most important structure and then specialize in distinct patterns.

The reviewers agreed that the paper was well-written, the task was a well-motivated simplification and in some way generalization of the higher-order markov chains tasks from prior work, and the analysis was novel and interesting. The main concerns were around (1) the simplifications to the architecture which made it feel more like a MLP-mixer than an attention module, specialized initializations, (2) over-claiming the role of sparse attention patterns for emergence and symmetry initialization for competition without showing causality, and (3) unclear connection to or evaluation on real NLP tasks. The reviewers found the author response to be satisfactory, and all eventually leaned toward accept.

I found the paper interesting with new insights that would be of interest to the folks working on training dynamics, therefore, I lean towards accept. I encourage the authors to take into account all the suggestions from the reviewers and update their manuscript to be more honest about their claims and the limitations of the analysis. Also, add the improved analysis that gets rid of their Assumption 2.